# LeanGeo: Formalizing Competitional Geometry problems in Lean

## Abstract

Geometry problems are a crucial testbed for AI reasoning capabilities. Most existing geometry solving systems cannot express problems within a unified framework, thus are difficult to integrate with other mathematical fields. Besides, since most geometric proofs rely on intuitive diagrams, verifying geometry problems is particularly challenging. To address these gaps, we introduce Lean-Geo, a unified formal system for formalizing and solving competition-level geometry problems within the Lean 4 theorem prover. LeanGeo features a comprehensive library of high-level geometric theorems with Lean's foundational logic, enabling rigorous proof verification and seamless integration with Mathlib. We also present LeanGeo-Bench, a formal geometry benchmark in Lean-Geo, comprising problems from the International Mathematical Olympiad (IMO) and other advanced sources. Our evaluation demonstrates the capabilities and limitations of state-of-the-art Large Language Models on this benchmark, highlighting the need for further advancements in automated geometric reasoning. To further improve prover performance, we introduce a synthetic data generation pipeline together with a reinforcement learning training framework built on LeanGeo. We open source the theorem library and the benchmark of LeanGeo at https://anonymous.4open.science/r/LeanGeo-9CE9

## 1 Introduction

In recent years, Large Language Models (LLMs) have made significant progress in mathematical reasoning, particularly in automated theorem proving (Bibel, 2013). Formal theorem proving is a crucial domain for ensuring the correctness of hard-to-verify proofs within theorem proving. Lean 4 (Moura & Ullrich, 2021), as a prominent proof assistant, provides a solid foundation for algebra and number theory through its extensive Mathlib library (mathlib community, 2020). It has been widely used in the formal verification of theorems within LLMs.

However, Euclidean geometry, an essential component of mathematical reasoning and a frequent focus of competitions, remains relatively underexplored in Lean 4 community, Mathlib and automated theorem provers. This stems from the inherent difficulty of geometric problems, which demand graphic intuition; human reasoning in such cases inevitably relies on geometric insight, making absolute formalization of geometry problem extremely challenging.

Currently, advanced geometric systems like AlphaGeometry (Trinh et al., 2024), TongGeometry (Zhang et al., 2024a) and SeedGeometry (Chen et al., 2025), while achieving impressive results on IMO-level geometry problems, typically rely on specialized models and operate within geometry-specific formal systems independent of Lean. This isolation prevents integration with other mathematical domains in mathlib, making it impossible to express geometric inequality positional relations. Additionally, their reliance on graphical verification and unordered formal systems can lead to logical unsoundness and inability to perform rigorous verification.(See detailed comparison in Table 3 and Appendix E.

Even in Lean 4, geometric results remain scarce: Mathlib's formalized geometry remains largely algebraic and provides little support for synthetic reasoning. Myers (Zhang et al., 2022) has formalized a single IMO geometry problem in Lean—an impressive isolated result—but the proof is written in a highly technical Mathlib-specific style and does not develop any structured or reusable geometric library, leaving the broader landscape of synthetic geometry in Lean essentially empty.

While developing a robust formal system is a vital step toward rigorous geometric reasoning, equally important is the establishment of suitable benchmarks to rigorously evaluate the geometric reasoning capability of LLMs. However, since most geometric proofs rely on intuitive diagrams, verifying geometry problems is particularly challenging. Existing geometry benchmarks, such as Geoeval (Zhang et al., 2024b), GeoQA (Chen et al., 2021) Geometry3K (Hiyouga, 2025) and Formal-Math (Yu et al., 2025), primarily emphasize numerical computations of geometry object, focusing on models' computational ability rather than their true geometric reasoning skills. Currently, LLMs exhibit unsatisfactory performance on Lean4 geometry benchmark such as MATP-bench (He et al., 2025) due to the absence of a geometry theorem library as tools to prove theorems. This highlights the necessity of developing a complete formal system and an extensive theorem library to serve as reliable tools for LLMs.

To handle these critical gaps, we introduce LeanGeo, a framework designed to formalize and solve geometric problems in Lean 4. Building upon LeanEuclid (Murphy et al., 2024), LeanGeo establishes a comprehensive library of geometric theorems specifically curated for competition-level challenges and seamlessly integrates with Mathlib. Compared to other formal systems like Alpha-Geometry, LeanGeo exhibits significant differences, as detailed in Table 1.

Table 1: Comparison of problem with AlphaGeometry and LeanGeo

| Natural Language | In a triangle $ABC$, side $AB = AC$, prove that $\angle ACB = \angle ABC$.

**Solution.** Choose $D$ as the midpoint of side $BC$. Then $\triangle ABD$ and $\triangle ACD$ are congruent. Therefore, $\angle ACB = \angle ACD = \angle ABD = \angle ABC$ |
|---|---|
| AlphaGeometry | a b = segment a b; c = on_circle c a b ? eqangle b a b c b c c a

**Solution.** * From theorem premises:
A B C : Points
cong A C A B [00]
* Auxiliary Constructions:
: Points
* Proof steps:
001. cong A C A B [00] $\Rightarrow$ eqangle A C B C B C A B |
| LeanGeo | theorem isoTriangle_imp_eq_angles : $\forall$ (A B C : Point),
IsoTriangle A B C $\rightarrow \angle$ A:B:C $=\angle$ A:C:B := by
  euclid_intros
  euclid_apply exists_midpoint B C as D
  euclid_apply line_from_points B C as BC
  euclid_apply coll_angles_eq
  euclid_apply congruentTriangles_SSS D B A D C A
  euclid_apply coll_angles_eq
  euclid_finish |

Based on this theorem library, we propose LeanGeo-Bench, the first formalized geometric problem benchmark in Lean 4. It comprises 122 geometry problems, including all International Mathematical Olympiad (IMO) geometry problems since 2000. Furthermore, we present a training methodology that uses the theorem library to construct supervised fine-tuning (SFT) data. This data is then used in reinforcement learning (RL) experiments upon the Kimi k1.5 reinforcement learning (RL) pipeline (Team et al., 2025), yielding promising initial results.

The primary contributions of this work are as follows:

- We present the first framework in the Lean theorem prover capable of expressing and reasoning about competition-level geometry problems in a human-like manner. The framework features an extensive library of high-level definitions and tactics based on theorems commonly used by IMO competitors, making formal proofs more intuitive and understandable. Its integration within Lean facilitates the formalization of problems at the intersection of geometry and other domains like combinatorics.

- We introduce a comprehensive geometry benchmark formalized in Lean 4 and LeanGeo, capable of representing most of the geometry problems from the International Mathemati-

cal Olympiad (IMO). This benchmark provides a standardized and challenging testbed for evaluating future formal mathematics systems. We also provide baseline results on this benchmark using several state-of-the-art large language models.

- We develop a novel method to generate synthetic data for competitional geometry problems and a Reinforcemnet Learning pipeline to instill unseen knowledge for LLMs.

## 2 RELATED WORK

### 2.1 AUTOMATED THEOREM PROVING

Interactive theorem provers span a spectrum of foundational languages: HOL4 (Slind & Norrish, 2008) and Isabelle/HOL (Paulson, 1994) rely on simply-typed higher-order logic, Coq (Barras et al., 1999) and Lean (De Moura et al., 2015) on dependent type theory.

In parallel, a series of search-based theorem provers have been developed to enhance automated reasoning capabilities. LEGO-Prover (Wang et al., 2023a) employs a modular formal proof framework to construct a reusable skill library, enabling LLMs to retrieve existing skills and synthesise new ones during the proof process. DT-Solver (Wang et al., 2023b) introduces a dynamic-tree Monte Carlo search algorithm, whereas BFS-Prover (Xin et al., 2025), based on a best-first search strategy, achieves state-of-the-art performance among search-based theorem provers.

More recent developments have shifted towards an alternative whole-proof generation approach, where a language model generates the entire proof in a single pass. Notable examples following this paradigm include DeepSeek-Prover (Ren et al., 2025), Goedel-Prover (Lin et al., 2025), and Kimina-Prover Preview (Wang et al., 2025). Agentic methods such as Delta Prover(Zhou et al., 2025) integrate reflective decomposition and iterative repair, allowing a general-purpose LLM to interactively construct formal proofs. Seed-Prover (Chen et al., 2025) combines multi-stage reinforcement learning, agent-based strategies and test-time scaling, achieving impressive results by fully solving 4 out of 6 problems in IMO 2025.

### 2.2 LEANEUCLID

LeanEuclid (Murphy et al., 2024) represents a pioneering effort in formalizing plane geometry within Lean by integrating SMT (Barrett & Tinelli, 2018) solving techniques with SystemE (Avigad et al., 2009) to construct a rigorous axiomatic framework. It introduces an autoformalization benchmark that covers the first chapter of Euclid's *Elements* along with 125 relatively simple problems drawn from the UniGeo corpus.

Table 2: Comparison between LeanEuclid and LeanGeo

|                              | LEANEUCLID       | LEANGEO                |
|------------------------------|------------------|------------------------|
| Axiom Number                 | 107              | 116                    |
| Theorem Number               | 106              | 260                    |
| Geometry Structure Number    | 12               | 50                     |
| Average Proof Length         | 20.27            | 16.20                  |
| Average number of quote lemma| 3.80             | 3.43                   |
| SMT Method                   | Hard-coded rules | LeanSMT                |
| Level                        | Euclid's Element | Competitional Geometry |

Our framework LeanGeo is a substantial expansion of LeanEuclid's theorem library and geometric structures. LeanEuclid formalizes only the 49 propositions in Elements I; as a result, its expressive power is far from adequate for solving standard middle- and high-school geometry problems. LeanGeo builds on the same axiomatic foundation but provides a significantly richer collection of theorems, definitions, and geometric structures while improving SMT method. A summary comparison is shown in Table 2.

## 2.3 GEOMETRY PROBLEM SOLVING

Automatic geometry solvers have a rich history. Classical algebraic methods—Wu's characteristic set (Wu, 1986) and Gröbner bases (Bose, 1995)—reduce geometry to polynomial ideal membership, achieving impressive coverage of textbook theorems.

A recent milestone in automated geometry reasoning is AlphaGeometry (Trinh et al., 2024), which integrates a neural language model trained on 100 million synthetic theorems with a symbolic deduction engine to solve 25 out of 30 IMO-level problems. Building on the framework proposed in Chou et al. (2000), its formal system is unordered and point-centered, enabling fast symbolic deduction within this setting. However, this formal system also has several notable limitations that restrict its broader applicability.

In essence, AlphaGeometry functions as a task-specialized solving system tailored for IMO-style geometry problems: it is extremely powerful in problem solving, but this comes at the cost of sacrificing internal axiomatic rigor and omitting several components we believe are equally essential for geometry learners and researchers—such as geometric inequalities, trigonometric reasoning, and positional or incidence relations. Furthermore, its unsound formal system makes it impossible to formally verify any proofs. While its simplified formal system accelerates search and inference, it loses part of the rigor and human interpretability. In contrast, our system aims to be more complete, rigorous, and structurally expressive, though this naturally results in more intricate and elaborate reasoning processes.

The comparison between LeanGeo and AlphaGeometry are shown in Table 3. Appendix E Gives more example to illustrate the comparison in the table.

Table 3: Comparison between AlphaGeometry and LeanGeo

| Category | Feature | AlphaGeometry | LeanGeo |
|---|---|---|---|
| Expressivity | Geometric Inequality & Trigonometric Functions | × | ✓ |
| | Metric Relation (Perpendicular, Parallel, Equal) | ✓ | ✓ |
| | Positional Relation (Inside, Between, Sameside) | × | ✓ |
| | Existential Proposition | × | ✓ |
| | Linear Computation | ✓ | ✓ |
| | Non-linear Computation | × | ✓ |
| Verifiability | Verifiability of Proof | × | ✓ |
| Axiom System | Soundness | × | ✓ |
| | Extensibility | × | ✓ |

## 2.4 GEOMETRY AND LEAN BENCHMARKS

Advances in automated theorem proving have spurred the development of various Lean-based mathematical benchmarks in recent years. MiniF2F, for instance, is a benchmark designed to evaluate automated theorem-proving systems on high-school-level algebra and number theory problems.

In parallel, several geometry benchmarks have been established to assess the multi-modal reasoning capabilities of large language models (LLMs). Benchmarks such as Geoeval (Zhang et al., 2024b), GeoQA (Chen et al., 2021), Geometry3K (Hiyouga, 2025), and FormalMath (Yu et al., 2025) offer comprehensive evaluations of computational and quantitative reasoning. However, classical geometric proof—rooted in Euclidean tradition—remains an essential aspect of geometric reasoning that is currently underrepresented in existing benchmarks, largely due to the difficulty of formal verification. LeanEuclid, built upon Book I of Euclid's *Elements*, provides a benchmark for autoformalization, yet its problem set is limited in scope and primarily consists of elementary exercises. The AlphaGeometry framework introduced two benchmarks, IMO-30 and JGEX-231, but these emphasize problem-solving without supporting verifiable formal proofs due to limitations in their underlying reasoning systems. MATP aggregates a large set of geometry problems written in Lean4, yet current LLMs perform unsatisfactorily on this benchmark. Moreover, the lack of a comprehensive geometry theorem library in Lean4 hinders the effective application of geometric tools by LLMs in this formal environment. A detailed comparison of these benchmarks is provided in Table 4.

Table 4: Comparison of Geometry and Lean Benchmarks

| Benchmark | Size | Verifiable | Geometric Formal Proving Percentage | Lean | Theorem Library | Level |
|---|---|---|---|---|---|---|
| miniF2F | 488 | ✓ | 0% | ✓ | Mathlib | Middle School |
| Geometry3K-test | 601 | ✓ | 0% | × | × | Middle School |
| LeanEuclid | 173 | ✓ | 0% | ✓ | SystemE | Elementary |
| AG-IMO-30 | 30 | × | 100% | × | DD rules | Olympiad |
| MATP-Bench | 1056 | ✓ | About 20% | ✓ | × | Synthetic |
| **LeanGeo-Bench** | 123 | ✓ | 100% | ✓ | LeanGeo | Synthetic |

## 3 LEANGEO

LeanGeo is a manually formalized system of plane geometry theorems and their proofs in the Lean 4 proof assistant. It builds upon the axiomatic framework of SystemE (Avigad et al., 2009), while its implementation inherits most foundational geometric objects, relations from LeanEuclid (Murphy et al., 2024), with slight modifications (see Appendix C). Additionally, LeanGeo leverages LeanSMT (Mohamed et al., 2025) at its core, which effectively hides many of the underlying proof details in Lean 4.

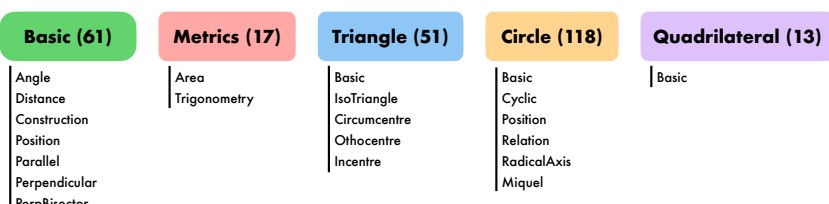

Figure 1: Structure of LeanGeo Theorem Library

### 3.1 THEOREM LIBRARY

To enhance the expressive power of the theorem library and align it with common geometric terminology, we firstly introduced 52 new definitions for geometric structures — such as Midpoint, Circumcenter, and RadicalAxis using `abbrev` as shown in 1. These additions make problem statements more concise and proofs more streamlined, while not increasing the length of the corresponding SMT process.

```
abbrev Cyclic (A B C D: Point) : Prop :=
∃ (O: Circle), A.onCircle O ∧ B.onCircle O ∧ C.onCircle O ∧ D.onCircle O
```

Listing 1: Example of abbreviation

With the assistance of these newly defined structures, we established LeanGeo, a theorem library comprising 260 geometric theorems as shown in 1. All theorems in the library are manually written, formally proved and auto-verified by Lean4 and LeanSMT.

These theorems systematically cover topics ranging from foundational middle-school geometry to challenging International Mathematical Olympiad (IMO) level theorem, such as Menelaus's theorem and Miquel's theorem. Besides, the library covers a wide range of geometry theorem, including fundamental properties of triangles (e.g., congruence, similarity), circles (e.g., inscribed angles, power of a point, radical axis), and quadrilaterals, as well as theorems related to key geometric points like the circumcenter and orthocenter.

A key feature of LeanGeo is that most proofs in the library are constructed by referencing previously established theorems through the `euclid_apply` tactic. Consequently, the development of the library parallels the human process of building geometric theory—progressing from axioms and simple foundations to increasingly complex structures (see Listing 6). As the library grows, these reusable lemmas substantially enhance deductive efficiency and shorten higher-level proofs.

Our experiments (See details in Appendix A.2) why this modular structure matters: integrating lemmas directly back into a theorem increases compilation time, and the effect becomes severe when lemma granularity is too coarse, as the system is forced to repeatedly recompile the same reasoning steps. In contrast, keeping lemmas separate allows shared arguments to be compiled once and reused, significantly improving overall efficiency.

Besides, LeanGeo is designed for seamless integration with Mathlib, enabling it to leverage powerful tools from other areas of mathematics. For example, it can employ trigonometric identities and advanced inequalities to tackle problems that are often beyond the reach of purely axiomatic geometry systems. As shown in D.2, trigonometric theorems in Mathlib are applied to prove IMO_2001_P1, a geometry inequality problem that is difficult to express within most geometric formal systems.

One of the most challenging issues in theorem annotation is describing positional relationships in geometry without visual aids. For problem illustrated in Figure 2, natural language proofs, as well as most geometry formal systems such as AlphaGeometry, consider only a single case. Owing to Lean's stringent requirements for rigor, a LeanGeo-proof must explicitly account for all possible cases. While this often results in more intricate proofs, it also ensures a higher level of rigor.

---

**Natural Language:**
The Circle O1 and O2 intersects at K and B, A line through B intersects the circle O1 at A and circle O2 at C. Prove that triangle KO1O2 and KAC are similar.

---

**Formal Statement in LeanGeo:**
theorem intersectCircles_similarTriangles_of_one_secant : $\forall$ (O₁ O₂ A B C K : Point) (Ω₁ Ω₂: Circle), Ω₁ $\neq$ Ω₂ $\wedge$ O₁.isCentre Ω₁ $\wedge$ O₂.isCentre Ω₂ $\wedge$ CirclesIntersectAtTwoPoints Ω₁ Ω₂ B K $\wedge$ A.onCircle Ω₁ $\wedge$ C.onCircle Ω₂ $\wedge$ Coll A B C $\wedge$ A $\neq$ B $\wedge$ B $\neq$ C $\wedge$ A $\neq$ K $\wedge$ C $\neq$ K $\rightarrow$ SimilarTriangles O₁ O₂ K A C K := by

---

**Possible Graph 1:**

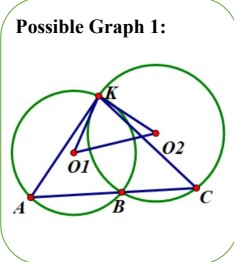

**Possible Graph 2:**

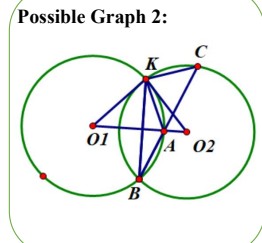

**Possible Graph 3:**

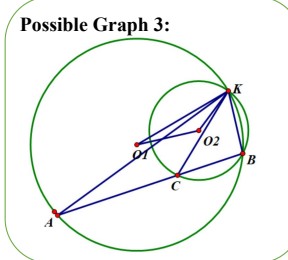

Figure 2: Different graphs with a same formal statement

To avoid overly cumbersome case analyses, we make extensive use of SMT solvers in our formal proofs to simplify the classification process and trivial results.

### 3.2 LEANSMT 4.15

To efficiently discharge goals deemed trivial in natural language proofs, LeanGeo invokes the CVC5 (Barbosa et al., 2022) SMT solver. In LeanEuclid (Murphy et al., 2024), the SystemE axioms are embedded as hardcoded SMT commands. By contrast, LeanGeo employs the `esmt` tactic, which directly passes all local hypotheses from the current tactic state—together with SystemE's inference axioms and the negated goal—to CVC5 for an unsatisfiability check. If CVC5 returns `unsat`, the entailment is confirmed.

For performance optimization, raw axiom expressions are not repeatedly translated into SMT commands. Instead, parsed axiom expressions are cached, and a global metavariable (mvar) dependency graph is maintained. This graph is dynamically updated whenever a definition or axiom annotated with `@[euclid]` is encountered as shown in Listing 2. The core logic for updating this dependency graph is presented in the Appendix B.

```
@[euclid]
axiom zero_segment_if :
  ∀ (a b : Point),  |(a - b)| = 0 → a = b
```

Listing 2: tactic usage]Example of @[euclid] tactic usage

The `@[euclid]` tactic makes our system more extensible. In LeanEuclid, the translator does not natively handle new definitions, meaning it would require manual modification to work with non-SystemE definitions such as sin and cos. Our system is designed to seamlessly incorporate such new definitions, making it more adaptable to a wider range of geometric problems. In addition, our theorem library inherits the expression styles of other tactics from LeanEuclid, such as `euclid_intros`, `euclid_apply`, and `euclid_finish`. When these tactics are executed, the system automatically invokes LeanSMT to return the results. The specific usage and examples of these tactics can be found in Appendix D.1.

Moreover, we analyze the scalability of LeanGeo based on four controlled experiments that vary the number of geometry elements, assumptions, proof length, and uses of `euclid` tactics (see Appendix A.1 for detailed graphs). Across all settings, both heartbeats and compilation time exhibit nearly linear growth with respect to the problem size, and the two metrics remain strongly positively correlated. increases mildly, but without causing instability. The four scaling curves demonstrate that LeanGeo's performance is dominated by the expected linear relationship between proof workload and compilation effort, with no pathological slowdowns observed.

## 4 LEANGEO-BENCH

### 4.1 BENCHMARK

LeanGeo-Bench is a formal benchmark tailored for formalizing and proving contest-level plane geometry theorems in Lean 4 and LeanGeo. As shown in Table 5, the benchmark consists of 122 problems drawn from diverse sources, including existing theorem libraries, textbooks, synthetically generated problems, contest problems.

Table 5: Composition of LeanGeo-Bench

| SECTION | N | SOURCE | METHOD |
|---|---|---|---|
| UniGeo(UG) | 10 | LeanEuclid | Manually Written |
| Library(LB) | 10 | LeanGeo Library | Manually Written |
| Synthetic Problem(SP) | 20 | LeanGeo Library | Generated by gemini |
| High Shool Competition(HSC) | 20 | NuminaMath | Autoformalized + double check |
| Olympic Problem(OP) | 19 | Evan Chen's textbook | Autoformalized + double check |
| IMO | 43 | AoPS | Autoformalized + double check |

The benchmark's difficulty ranges from foundational to competition-level. It includes 20 introductory problems: 10 from UniGeo(Chen et al., 2022) and 10 from LeanGeo theorem library. Another 20 problems ('Gemini_synthetic') are synthetically generated by an gemini-2.5 via our Problem Generation Pipeline. The majority of the benchmark consists of 83 more advanced problems sourced from high-school curricula, NuminaMath(Li et al., 2024), Evan Chen's Geometry textbook Chen (2021), and all the International Mathematical Olympiad (IMO) geometry problems since 2000 from AoPS(Art of Problem Solving). These problems were developed using a human-in-the-loop methodology: For each problem, it is first autoformalized by a large language model through prompt engineering, and then rigorously reviewed and corrected by two human experts.

The benchmark covers a broad range of topics commonly encountered in competitive geometry, including triangles, circles, quadrilaterals, and notably triangle centers (e.g., incenter, circumcenter), as shown in Figure 3. It also contains comprehensive problems involving multiple geometric configurations. Moreover, the problem types are diverse: in addition to traditional plane geometry proofs, many problems require calculating or deriving angles and side lengths. The benchmark further includes three geometry inequality problems and two problems involving moving points.

As part of this work, we present 43 formally verified solutions to problems in the benchmark, including two from the International Mathematical Olympiad (IMO), all of which are machine-checked in Lean. The formal proofs ensure the correctness of these problems. For problems without formal

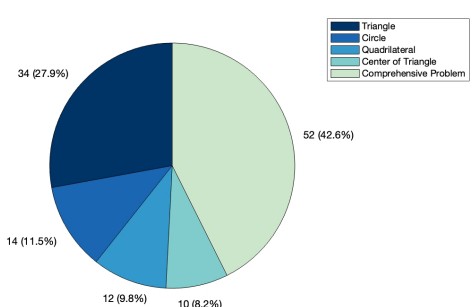

Figure 3: Category Distribution of LeanGeo-Bench

proofs, we validate correctness using a negation-based method combined with independent reviews by two geometry experts.

## 4.2 EVALUATION METHOD

To guide the LLM in generating formal proofs, we design a comprehensive prompt that carefully structures the task environment. The prompt comprises a custom declarative Domain-Specific Language of LeanGeo, "Error-and-correction" examples, construction rules for geometric definitions, the full set of theorems from the LeanGeo theorem library, together with few-shot learning examples. The complete prompt is provided in the Appendix F.

To evaluate the result generated by LLM, we apply the `online_one_stage` Fine-Eval method introduced in CombiBench (Liu et al., 2025) - This evaluation followed a two-step procedure. First, we checked that the LLM's result was consistent with the initial formal problem statement. Then, we fed the result into a Lean server containing a pre-built theorem library to formally verify the proof.

## 4.3 BASELINE RESULT

To comprehensively evaluate the model's performance on the benchmark, we conducted extensive testing across Gemini 2.5 Pro (DeepMind, 2025), o4-mini (OpenAI, 2025), Grok 4 (xAI, 2025), Kimi K2 (MoonshotAI, 2025), Claude 4 (Anthropic, 2025) and Qwen3-235B-A22B (Yang et al., 2025) and collected their overall success rates at different sample budgets and their performance in different section. The results are shown in Table 6.

Table 6: Evaluation on LeanGeo-Bench

| MODEL | OVERALL SUCCESS RATE (%) | | | SUCCESS NUMBER(pass@4) | | | | | |
|---|---|---|---|---|---|---|---|---|---|
| | pass@1 | pass@2 | pass@4 | UG | LB | SP | HSC | OP | IMO |
| Gemini 2.5 Pro | 17.21 | **22.95** | **27.05** | **10** | 4 | **13** | **6** | 0 | 0 |
| o4-mini | **19.67** | 21.31 | 22.13 | 7 | **9** | 8 | 3 | 0 | 0 |
| Grok 4 | 16.39 | 21.31 | 24.59 | **10** | 6 | 11 | 3 | 0 | 0 |
| Kimi K2 | 9.02 | 9.02 | 9.84 | 1 | **9** | 2 | 0 | 0 | 0 |
| Claude 4 | 4.92 | 9.02 | 10.66 | 1 | 5 | 7 | 0 | 0 | 0 |
| Qwen3-235B-A22B | 3.28 | 4.10 | 5.74 | 0 | 6 | 1 | 0 | 0 | 0 |
| | | | Total | 10 | 10 | 20 | 20 | 19 | 43 |

The LeanGeo-Bench results reveal substantial differences in geometric theorem-proving performance across state-of-the-art LLMs. o4-mini (OpenAI, 2025) attains the highest pass@1 score (19.67%), while Gemini 2.5 Pro (DeepMind, 2025) leads at pass@4 (27.05%).

A breakdown by category at pass@4 reveals complementary strength of LLMs in different area: Gemini-2.5-Pro excels in novel-problem settings such as Synthetic Proof (SP) and High School Competition (HSC), indicating stronger adaptability to unseen reasoning patterns, while GPT-o4-mini demonstrates greater proficiency in Library(LB), suggesting a more understanding and application of the theorem library in prompt.

While most models achieve partial success on the benchmark, their performance plateaus below 30%, and notably none of the evaluated models could solve any of the 62 Olympic-level problems, indicating fundamental limitations in handling complex geometric proofs that require sophisticated logical reasoning, advanced diagram interpretation, and formal verification capabilities.

## 5 REINFORCEMENT LEARNING EXPERIMENTS

### 5.1 GENERATING DATA BY LLM

A significant challenge in applying Reinforcement Learning training on LeanGeo is the absence of pre-existing cold start data, as LeanGeo establishes a novel framework for formal geometry. To address this, we developed a synthetic data generation pipeline. This process begins by creating a specialized prompt for Gemini 2.5 Pro (DeepMind, 2025), featuring carefully crafted guidelines and few-shot examples of theorem generation. Instead of tasking the LLM with solving a predefined problem, we prompt it with five randomly sampled theorems from our existing LeanGeo library. The LLM is then instructed to synthesize a new theorem and a corresponding proof, using the sampled theorems as inspiration. We repeated this process 5,000 times, each time conditioning the model on a different random subset of our library, to ensure a broad and diverse distribution of new problems.

The generated theorem-proof pairs are then automatically verified using the Lean prover. This verification reveals that 89% of the generated formal statements are syntactically valid, and 14% of the full submissions (statement and proof) pass the verification. Based on this outcome, we categorize the generated data: the activation dataset consists of problems with a valid statement and correct proof. This dataset is used for supervised fine-tuning as the initialization phase for reinforcement learning, while problems with valid statement but invalid proof are used for the prompt set in reinforced learning. The whole process is illustrated in Figure 4.

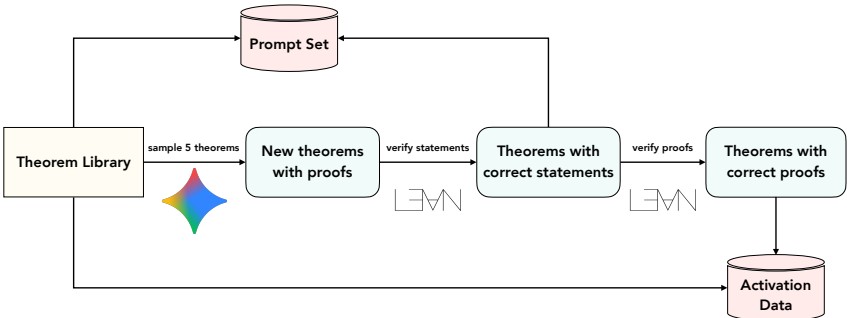

Figure 4: Data Generation Pipeline

### 5.2 INSTILLING KNOWLEDGE IN RL

Another challenge arises from the size of our theorem library. To prove a new theorem, the model must select and apply relevant theorems from this library. Incorporating the entire library into the prompt may present practical limitations, as it risks surpassing the model's context window, which could adversely impact training efficiency and model performance. To overcome this, we propose an "instilling method" that structures the prompt to manage the context effectively. Specifically, we use the following data format:

```
You are an expert in Lean 4 and geometric problem-solving.
You may apply the following theorems to solve the problem:
<theorem_1>
<theorem_2>
...
<theorem_10>
Now, let's solve the following problem step-by-step.
<formal_statement>
```

During reinforcement learning, we retain the same prompt structure; however, the 10 provided theorems are selected entirely at random from the library, regardless of their relevance to the target

formal statement. This approach encourages the model to discern and apply theorems that are truly pertinent within a noisy context, fostering a critical skill necessary for effective problem-solving.

## 5.3 RL Training

We employ the RL framework of the Kimina-Prover (Wang et al., 2025) to train our model. Our RL training procedure consists of two stages. Initially, the agent is trained on the activation dataset, during which the model's proof success rate improves from a post-SFT baseline of 37% to 60%. Subsequently, training proceeds on the prompt set, where the success rate increases from 12.5% to 40%. This training regimen also yields enhanced performance on our evaluation benchmark, with the pass@1 rate rising from 2.52 % to 10.92%.

## 6 Discussion and Future Work

While LeanGeo successfully demonstrates the viability of a declarative, human-readable approach for competition-level geometry, several key challenges and opportunities for future work remain. These are centered on strengthening the system's foundational soundness, enhancing its automation capabilities, and Instilling domain-specific knowledge to LLMs.

### 6.1 Automation Capabilities

While the integration with SMT solvers is powerful, a limitation of general-purpose SMT solvers is their lack of geometry-specific heuristics. Therefore, the solving speed of SMT significantly decreases as the number of points in the problem increases. One way to scale LeanGeo for more complex problems is by embedding domain-specific proof automation, like the Area Method(Janicic et al., 2012) or algebraic geometry techniques, into the tactic framework.

### 6.2 Instilling Domain-Specific Knowledge to LLMs

In the current benchmark, to ensure the model correctly cites theorems, we input the entire theorem library's statements as prompts to the model. However, long prompts may negatively impact the model's performance.

To address this issue, our RL framework takes first steps in reducing prompt length and instilling knowledge into LLMs. However, our method is still rather rudimentary and needs more sophisticated development.

## 7 Conclusion

In this paper, we present LeanGeo, the first Lean-based framework capable of formalizing and solving competition-level geometry problems, together with LeanGeo-Bench, a 122-problem benchmark spanning from foundational theorems to IMO challenges. LeanGeo's declarative, human-readable proofs, deep Mathlib integration, and extensible library enable rigorous cross-domain reasoning beyond the reach of existing geometry systems.

Our baseline evaluations reveal that while current LLMs can solve some problems, they fall far short on the hardest tasks, underscoring the need for stronger geometric reasoning and proof search capabilities. By combining a rich formal library, a challenging benchmark, and initial reinforcement learning experiments, LeanGeo establishes a scalable testbed for advancing automated geometry theorem proving and neuro-symbolic reasoning.

## 8 Reproducibility Statement

To reproduce the LeanGeo experiments or run the benchmark evaluation reported in this paper, please clone the anonymized repository at `https://anonymous.4open.science/r/LeanGeo-9CE9` and follow the step-by-step instructions given in the README.md. Our evaluation toolkit offers a clean, end-to-end benchmark harness: one command clones the repo, downloads frozen artifacts, and prints the identical numbers reported in the paper—no manual tuning or secret flags—thereby maximizing reproducibility. The RL-training pipeline relies on Moonshot AI internal infrastructure that cannot be released.

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

# A ANALYSIS OF SCALABILITY

## A.1 SCALABILITY OF SMT

We conduct a series of supplementary experiments to evaluate the scalability of LeanGeo, examining how four key factors influence both compilation time and the number of heartbeats required for proof execution:

- [(1)] The number of basic geometric elements (points, lines, and circles),
- [(2)] The number of given conditions,
- [(3)] The length of the proof, and
- [(4)] The number of applications of the `euclid` tactics.

The scaling curves are shown in Figure 5.

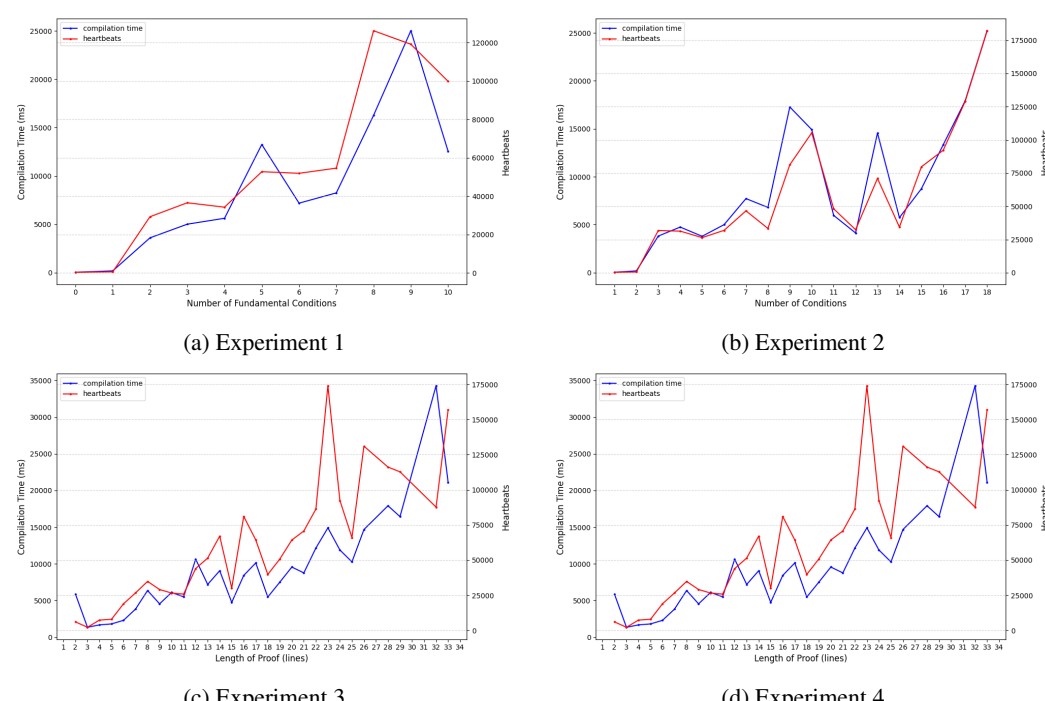

(a) Experiment 1

(b) Experiment 2

(c) Experiment 3

(d) Experiment 4

Figure 5: Scaling behavior of heartbeats and compilation time across four experimental settings.

Across all four experiments, LeanGeo exhibits approximately linear scaling: as we increase assumptions, conditions, proof length, or the number of Euclid tactics, both heartbeats and compilation time grow in a strongly correlated, near-linear manner. The only noticeable rises occur when the logical structure becomes denser (e.g., deeper lemma dependencies), which naturally increases the amount of proof search. Overall, the results show that LeanGeo is practically scalable, with performance determined primarily by the expected positive correlation between heartbeats and compilation effort rather than by any pathological geometric cases.

## A.2 COMPLEXITY VERSUS LEMMA GRANULARITY

Our experiments demonstrate that coarse lemma granularity leads to severe blow-ups in both compilation time and heartbeats. When large "all-in-one" lemmas are inlined directly into a theorem, many nearly identical reasoning steps must be recompiled repeatedly, causing exponential-like scaling.

In contrast, extracting commonly reused intermediate results into separate lemmas keeps the compilation cost close to linear in the dependency depth, because each lemma is compiled once and then reused. This is precisely why Lean's modular proof structure is essential for scalability.

The Table 7 illustrates this effect using Miquel's Theorem at different lemma-dependency depths:

| Lemma Depth | Compiled One Time | | Compiled Multiple Times | |
|---|---|---|---|---|
| | Heartbeats | Time(ms) | Heartbeats | Time(ms) |
| 0 | $1.7 \times 10^5$ | $2.1 \times 10^4$ | $1.7 \times 10^5$ | $2.1 \times 10^4$ |
| 1 | $6.3 \times 10^5$ | $8.8 \times 10^4$ | $7.6 \times 10^5$ | $1.0 \times 10^5$ |
| 2 | $1.4 \times 10^6$ | $2.1 \times 10^5$ | $2.7 \times 10^6$ | $4.0 \times 10^5$ |
| 3 | $2.5 \times 10^6$ | $3.6 \times 10^5$ | $8.9 \times 10^6$ | $1.3 \times 10^6$ |
| 5 | $4.6 \times 10^6$ | $7.5 \times 10^5$ | $1.0 \times 10^8$ | $1.8 \times 10^7$ |
| 8 | $7.5 \times 10^6$ | $1.3 \times 10^6$ | $1.4 \times 10^9$ | $2.2 \times 10^8$ |

Table 7: Scaling behavior of heartbeats and compilation time under different lemma depths.

In the table, "lemma depth" refers to the depth of dependencies referenced back from the current theorem, where a "lemma depth" of 0 indicates the current theorem itself. The right side represents the total compilation resource consumption at that lemma depth. If intermediate theorems are not extracted, a single theorem may need to be written and compiled multiple times. Conversely, extracting them ensures that the intermediate result is compiled only once.

## B  COMMAND CACHING

```
/--
Adds a command for a new constant to the SMT command cache and updates
   the dependency graph.

* `oldAxiomExprs`: the expressions corresponding to the types of all
   currently cached axioms.
* `cName`: the name of the axiom to be added to the cache.
* `initialState`: the current state of the global dependency graph.

Returns a tuple of the form `(new global dependency graph, new list of
   cached axioms, list of SMT commands for all of the axioms)`.
-/
def addCommandForConstant
  (oldAxiomExprs : List Expr)
  (cName : Name)
  (initialState : QueryBuilderM.State)
  : MetaM (QueryBuilderM.State × List Expr × List Command) := do
  let constInfo ← getConstInfo cName
  let constExpr := mkConst cName (constInfo.levelParams.map Level.param)
  let ((_, st), r) ←
    QueryBuilderM.buildDependencyGraph (mkConst `True)
    |>.run { toDefine := oldAxiomExprs ++ [constExpr] :
    QueryBuilderM.Config }
    |>.run initialState
    |>.run { uniqueFVarNames := {} : TranslationM.State }
  let (_, cmds) ← StateT.run (st.graph.orderedDfs (oldAxiomExprs ++
    [constExpr]) (emitVertex st.commands)) []
  return ⟨st, oldAxiomExprs ++ [constExpr], cmds⟩
```

Figure 6: Command caching code for SystemE axioms.

## C  CHANGES TO SYSTEME FORMALISM

There are some descrepencies between how SystemE axioms are described in the LeanEuclid lean theory vs how they are passed into the SMT solver. In particular degree and length and area are defined directly as functions from Points to a real number. That is the types Angle and

Segment do not exist in the SMT query. If a rule involves substituting a function into application into a forall statement it will double the search depth required to obtain that proof. For example if angle degree is defined as `Angle.degree (Angle.ofPoints a b c)` the smt's search procedure would have to first apply Angle.ofPoints to points $a, b, c$ and then apply Angle.degree to that resultant angle. By contrast, if degree is defined as the measure of three points only a single application is required to obtain the term `degree a b c`. By changing the definition of degree to be a function on three points it halves the search depth required to acheive the same term. Since we generally never reason about segments or angles outside of their measures this simplification is acceptable and segment congruence is defined uniquely by length. For Triangles it is not possible to get rid of the type entirely since Triangle congruence. We can however define a function `area'` which behaves as an area function on points. When then define `Triangle.area (Triangle.ofPoints a b c) = area' a b c`. And tag it as a simp lemma. Thus, since simplification is applied before passing into the smt solver, the Triangle type will dissappear by the time the smt solver is invoked. A similar trick can be done Triangle.congruence.

```
opaque Angle : Point → Point → Point → ℝ
-- ...
notation:71 "∠" a ":" b ":" c:72 => Angle a b c
```

Listing 3: Angle Definition

```
opaque area' : Point → Point → Point → ℝ

inductive Triangle
| ofPoints (a b c : Point)

@[simp]
abbrev Triangle.area : Triangle → ℝ :=
  fun x =>
    match x with
    | ofPoints a b c => area' a b c

notation:max "△" a ":" b ":" c:66 => Triangle.ofPoints a b c

instance : Coe Triangle ℝ :=
  ⟨Triangle.area⟩
```

Listing 4: Triangle Definition

Besides, to broaden SystemE's applicability to the wider field of geometry, we add nine axioms to LeanGeo covering circles, triangles, similar triangles, and triangle areas, which cannot be derived within the original SystemE.

```
axiom triangle_area_foot :∀ (a b c d: Point) (BC: Line),b.onLine BC ∧
    c.onLine BC ∧ (Triangle a b c) ∧ Foot a d BC → (△ a:b:c).area = |
    (a-d)| * |(b-c)|/2

axiom threePoints_existCircle : ∀ (A B C : Point),
  Triangle A B C →
  ∃ (Ω : Circle),
    (A.onCircle Ω ∧ B.onCircle Ω ∧ C.onCircle Ω)

axiom exists_centre : ∀ (O: Circle), ∃ (C : Point), C.isCentre O

axiom rightAngle_eq_pi_div_two : ∟ = Real.pi / 2

axiom rightTriangle_sin : ∀ (A B C : Point), RightTriangle A B C →
    Real.sin (∠A:B:C) = |(A-C)| / |(B-C)|

axiom rightTriangle_cos : ∀ (A B C : Point), RightTriangle A B C →
    Real.cos (∠A:B:C) = |(A-B)| / |(B-C)|
```

```
axiom similar_AA : ∀ (A B C D E F : Point), Triangle A B C ∧ Triangle D
    E F ∧ ∠ A:B:C = ∠ D:E:F ∧ ∠ B:A:C = ∠ E:D:F → SimilarTriangles A B
    C D E F

axiom similar_SAS : ∀ (A B C D E F : Point), Triangle A B C ∧ Triangle D
    E F ∧ ∠ A:B:C = ∠ D:E:F ∧ |(A-B)| * |(E-F)| = |(B-C)| * |(D-E)| →
    SimilarTriangles A B C D E F

axiom similar_SSS : ∀ (A B C D E F : Point), Triangle A B C ∧ Triangle D
    E F ∧ |(A-B)| * |(E-F)| = |(B-C)| * |(D-E)| ∧ |(B-C)| * |(F-D)| = |(C-A)| * |
    (E-F)| → SimilarTriangles A B C D E F
```

Listing 5: Additional Axioms in LeanGeo

# D EXAMPLES OF FORMALIZATION

## D.1 EXAMPLES IN THEOREM LIBRARY

Here is a proof example from the LeanGeo theorem library.

```
theorem angle_lt_outsideCircle: ∀ (A B C D : Point) (AB : Line) (Ω :
    Circle), A.onCircle Ω ∧ B.onCircle Ω ∧ distinctPointsOnLine A B AB ∧
    C.onCircle Ω ∧ C ≠ A ∧ C ≠ B ∧ D.sameSide C AB ∧ ∠A:D:B < ∠ A:C:B
    → D.outsideCircle Ω := by
  euclid_intros
  have h1 : ¬ (D.onCircle Ω) := by
    by_contra
    euclid_apply cyclic_eqAngle A B C D AB Ω
    euclid_finish
  have h2: ¬ (D.insideCircle Ω):= by
    by_contra
    euclid_apply line_from_points A D as AD
    euclid_apply intersection_circle_line_extending_points Ω AD D A as E
    have h3: ∠ B:C:A = ∠ B:E:A := by
      euclid_apply cyclic_eqAngle A B C E AB Ω
      euclid_finish
    euclid_apply triangle_exteriorAngle E D B A
    have h4: ∠ A:E:B = ∠ D:E:B := by
      euclid_apply angle_between_transfer A D E B
      euclid_finish
    euclid_finish
  euclid_finish
```

Listing 6: Example of Theorem Library

LeanGeo proofs are structured to mirror the step-by-step, declarative style of traditional, natural-language geometry proofs. This design choice results in simple, readable proof scripts that are particularly amenable to machine learning techniques. The proof development relies on a small set of core tactics:

- `euclid_intros`
  This is an initialization tactic that begins the proof. It processes the theorem's statement, automatically introducing all universally quantified variables (e.g., 'A', 'B', 'C', 'D', 'Ω') and hypotheses (e.g., 'A.onCircle Ω', 'D.sameSide C AB') into the local proof context.

- `euclid_apply <rule> <args>`
  Given a rule <rule> with type of the form ∀(<args> : Types) ... P -> Q, this tactic attempts to prove premise P from the local proof and attempts to prove premise P from the local proof context using an SMT solver. If successful, propsition Q is added to the proof context.
  In this example, `euclid_apply cyclic_eqAngle A B C D AB` refers to the former theorem in the library(in Circle.lean)

```
       theorem cyclic_eqAngle: ∀ (A B C D: Point) (AB:Line) (Ω :
       Circle), distinctPointsOnLine A B AB ∧ C≠ A ∧ D ≠ A ∧ C ≠ B ∧
        D ≠ B ∧ A.onCircle Ω ∧ B.onCircle Ω ∧  C.onCircle Ω ∧
       D.onCircle Ω ∧ C.sameSide D AB → ∠ B:C:A = ∠ B:D:A := by ...
```

LeanGeo automatically checks whether all of the premises of `cyclic_eqAngle`, i.e.
`distinctPointsOnLine A B AB`, `C ≠ A`, `D ≠ A ...` are satisfied. If yes,
then its result,`∠ B:C:A = ∠ B:D:A` will be added in the proof context.

- `euclid_apply <rule> with <args> as <x, h>`
  A forward-reasoning tactic designed to apply theorems and construction rules. Given a
  rule, typically of the form `∀..., P → ∃ x, Q(x)`
  This tactic instantiates it with the provided arguments `<args>`. It then employs an SMT
  solver to automatically prove the premise 'P' using hypotheses from the local context. If
  successful, the tactic introduces the newly constructed object 'x' and its property 'Q(x)'
  (named 'h') into the context. This command streamlines geometric constructions and de-
  ductions by combining the application of a rule with the automated verification of its pre-
  conditions, making the proof script more declarative and readable.

- `euclid_finish`
  A terminal tactic that invokes an SMT solver to automatically prove the current goal using
  the set of available hypotheses in the local context. This tactic is effective for discharg-
  ing goals that are either direct assumptions or straightforward logical consequences of the
  premises, requiring minimal search from the solver.

- `have hP : P := by`
  A construct for structuring proofs by introducing an intermediate lemma 'P' (named 'hP').
  This allows a complex proof to be decomposed into a sequence of smaller, more manage-
  able sub-proofs. This methodology not only enhances the readability and maintainability
  of the proof script but also improves the SMT solver's performance by reducing its search
  space. The solver can tackle the smaller lemma in isolation and then utilize the proven
  result 'hP' in the main proof.

### D.2 FORMALIZATION OF IMO 2001 P1

Problem statement:

```
Let ABC be an acute-angled triangle with O as its circumcenter. Let P
    on line BC be the foot of the altitude from A. Assume that
    ∠BCA ≥ ∠ABC + 30°. Prove that ∠CAB + ∠COP < 90°.
```

Proof of LeanGeo:

```
import Mathlib
import SystemE
import LeanGeo
open LeanGeo Real
--Consider an acute-angled Triangle ABC. Let P be the Foot of the
    altitude of Triangle ABC issuing from the vertex A, and let O be
    the circumcenter of Triangle ABC. Assume that ∠C ≥ ∠B + 30°. Prove
    that ∠A + ∠COP < 90°.
--To Trigonometry.lean
--To Triangle.lean
set_option maxHeartbeats 0

theorem sin_inequality(B C : ℝ)
  (hB : 0 < B ∧ B < π) (hC : 0 < C ∧ C < π)
  (hC1 : C ≥ B + π/6) : 4 * sin B * cos C ≤ 1 := by
  rcases hB with ⟨hB1, hB2⟩
  rcases hC with ⟨hC11, hC22⟩
  have h1 : cos C ≤ cos (B + π / 6) := by
    have h2 : C ≥ B + π / 6 := hC1
    have h3 : C < π := by linarith [hC22]
```

```
972        have h4 : 0 < B + π / 6 := by
973          linarith [hB1, Real.pi_pos]
974        have h5 : B + π / 6 < π := by
975          nlinarith [hB2, hC11, hC22, Real.pi_pos]
976        have h6 : cos C ≤ cos (B + π / 6) := by
977          apply Real.cos_le_cos_of_nonneg_of_le_pi
978          all_goals
979            nlinarith [Real.pi_pos, hB1, hB2, hC11, hC22, Real.pi_pos]
980        linarith
981      have h2 : sin B * cos (B + π / 6) ≤ 1 / 4 := by
982        have h21 : cos (B + π / 6) = cos B * cos (π / 6) - sin B * sin (π /
983        6) := by
984          rw [Real.cos_add]
985        have h22 : cos (π / 6) = Real.sqrt 3 / 2 := by
986          rw [cos_pi_div_six]
987        have h23 : sin (π / 6) = 1 / 2 := by
988          rw [sin_pi_div_six]
989        have h24 : sin B * cos (B + π / 6) = (Real.sqrt 3 / 2) * sin B * cos
990        B - (1 / 2) * sin B ^ 2 := by
991          rw [h21, h22, h23]
992          ring_nf
993        have h25 : (Real.sqrt 3 / 2) * sin B * cos B - (1 / 2) * sin B ^ 2 ≤
994        1 / 4 := by
995          nlinarith [sq_nonneg (sin B - 1 / 2), sq_nonneg (cos B - Real.sqrt
996        3 / 2),
997              sq_nonneg (sin B ^ 2 - 1 / 4), sq_nonneg (sin B - Real.sqrt 3
998        / 2),
999              sq_nonneg (cos B ^ 2 - 1 / 4), sq_nonneg (cos B - 1 / 2),
1000             Real.sqrt_pos.mpr (by linarith : (0 : ℝ) < (3 : ℝ)),
1001             Real.sqrt_nonneg 3, Real.sq_sqrt (show (0 : ℝ) ≤ (3 : ℝ) by
1002       linarith),
1003             Real.sin_sq_add_cos_sq B, mul_nonneg (show 0 ≤ (0 : ℝ) by
1004       linarith) (show 0 ≤ (0 : ℝ) by linarith),
1005             Real.sin_pos_of_pos_of_lt_pi hB1 (by linarith : B < Real.pi)]
          linarith [h24, h25]
        have h3 : 0 < sin B := by
          apply sin_pos_of_pos_of_lt_pi
          all_goals linarith [hB1, hB2, Real.pi_pos]
        nlinarith [h1, h2, h3, Real.sin_sq_add_cos_sq B,
          Real.sin_sq_add_cos_sq C, Real.pi_pos]

theorem sin_range (A : ℝ) (hA : 0 < A ∧ A < π/2) : sin A < 1 ∧ sin A > 0
    := by
  have h1 : 0 < A := hA.1
  have h2 : A < π / 2 := hA.2
  have h3 : sin A < 1 := by
    have h4 : sin (π / 2) = 1 := by
      rw [sin_pi_div_two]
    have h5 : sin A < sin (π / 2) := by
      apply sin_lt_sin_of_lt_of_le_pi_div_two
      all_goals linarith [Real.pi_pos, Real.pi_gt_three, h1, h2]
    linarith [h4, h5]
  have h6 : sin A > 0 := by
    have h7 : sin (0 : ℝ) = 0 := by
      simp [Real.sin_zero]
    have h8 : sin (0 : ℝ) < sin A := by
      apply sin_lt_sin_of_lt_of_le_pi_div_two
      all_goals linarith [Real.pi_pos, Real.pi_gt_three, h1, h2]
    linarith [h7, h8]
  constructor
  · linarith [h3]
  · linarith [h6]
--To Triangle, Generated b
```

```
theorem IMO_2001_P1 :
  ∀ (A B C P O : Point) (AB BC CA : Line),
    formAcuteTriangle A B C AB BC CA ∧
    Foot A P BC ∧
    Circumcentre O A B C ∧
    ∠ A:C:B ≥ ∠ C:B:A + ∟/3 →
    ∠ B:A:C + ∠ C:O:P < ∟ := by
  euclid_intros
  euclid_apply rightAngle_eq_pi_div_two
  euclid_apply acuteTriangle_circumcentre_insideTriangle A B C O AB BC CA
  euclid_apply circle_from_points O B as Ω
  euclid_apply circumcentre_inscribedAngle_comp B C A O BC Ω
  have h0: 4 * sin (∠ B:A:C) * sin (∠A:B:C) * cos (∠A:C:B) < 1 := by
    have h1: 0 < ∠ A:B:C ∧ ∠ A:B:C < π := by
      euclid_finish
    have h2: 0 < ∠ A:C:B ∧ ∠ A:C:B < π := by
      euclid_finish
    have h3: (sin (∠ B:A:C) < 1) ∧ (sin (∠ B:A:C) > 0) := by
      euclid_apply sin_range (∠B:A:C)
      euclid_finish
    have h4: ∠ A:C:B ≥ ∠ C:B:A + π/6 := by
      euclid_finish
    have h5: 4 * sin (∠A:B:C) * cos (∠A:C:B) ≤ 1 := by
      euclid_apply sin_inequality (∠A:B:C) (∠A:C:B)
      euclid_finish
    nlinarith

  have h1: between B P C := by
    euclid_apply acuteTriangle_foot_between A B C P BC
    euclid_finish
  have h2: |(P-C)| < |(P-O)| := by
    have h3: |(P-C)| * |(P-C)|  < |(P-O)| * |(P-O)| := by
      have h4: |(O-C)| * |(O-C)| - |(O-P)| * |(O-P)| = |(P-B)| * |(P-C)|:= by
        euclid_apply ApolloniusTheorem_to_isoTriangle O B C P BC
        euclid_finish
      have h5: |(P-C)| = |(A-C)| * cos (∠ A:C:P) := by
        euclid_apply rightTriangle_cos P C A
        euclid_finish
      have h6:  |(A-C)| = 2 * |(O-C)| * sin (∠A:B:C) := by
        euclid_apply LawOfSines_radius B A C O
        euclid_finish
      have h7:  |(B-C)| = 2 * |(O-C)| * sin (∠B:A:C) := by
        euclid_apply LawOfSines_radius A B C O
        euclid_finish
      have h8: ∠A:C:P = ∠A:C:B := by
        euclid_apply coll_angles_eq B P C A
        euclid_finish
      have h9: |(P-C)| * |(B-C)| < |(O-C)| * |(O-C)| := by
        rw [h5, h6, h7,h8]
        have h10: (|(O-C)| * |(O-C)|) > 0 := by euclid_finish
        calc
          _ = (4 * sin (∠ B:A:C) * sin (∠A:B:C) * cos (∠A:C:B)) *
    (|(O-C)| * |(O-C)|) := by linarith
          _ < 1 * (|(O-C)| * |(O-C)|) := by euclid_finish
          _ = _ := by euclid_finish
      euclid_finish
    euclid_assert |(P-C)| > 0
    euclid_assert |(P-O)| > 0
    nlinarith
  euclid_assert Triangle O C P
  euclid_apply triangle_gt_side_gt_angle P C O
  have h_final: ∠ P:C:O = ∠ B:C:O := by
    euclid_apply coll_angles_eq B P C O
    euclid_finish
```

```
    euclid_finish
```

Listing 7: Proof of LeanGeo for IMO 2001 P1

A significant advantage of LeanGeo is its seamless integration with Mathlib's extensive mathematical library, enabling it to tackle a broader class of problems . This is particularly evident in its ability to formalize geometric inequalities, a domain where systems like AlphaGeometry face challenges due to their reliance on converting geometry into polynomial equations. The formalization of IMO 2001 P1, shown above, serves as a prime example. The proof strategy involves reducing the geometric inequality $\angle CAB + \angle COP < \frac{\pi}{2}$ to a trigonometric one: $4\sin(\angle ABC)\cos(\angle BCA) \leq 1$, derived from the condition $\angle BCA \geq \angle ABC + \frac{\pi}{6}$.

This trigonometric lemma, 'sin_inequality', is proven not by geometric tactics. Annotators could obtain the proof from a open-sourced formal prover, Kimina-Prover Wang et al. (2025). The main geometric proof, orchestrated by LeanGeo's 'euclid_....' tactics, then imports and applies this analytical result to complete the formalization. This hybrid approach, combining high-level geometric reasoning with deep analytical capabilities from Mathlib, demonstrates LeanGeo's power in unifying different mathematical domains to expand the scope of automated geometric theorem proving.

# E  COMPARISON WITH ALPHAGEOMETRY

## E.1  EXPRESSIVITY

Compared with LeanGeo, AlphaGeometry(Trinh et al., 2024) is built upon a significantly weaker axiomatic foundation. Its formal language cannot express many essential geometric notions, including:

1. inequality and quantitative relations,
2. positional relations (inside, outside, between, same side),
3. existential quantifiers and locus-type assertions,
4. trigonometric functions and general real-number computation,
5. ordered-angle semantics required for precise angular reasoning.

To quantify this gap, we analyzed all 260 theorems in the LeanGeo library and found that **56.% (148 theorems)** are completely inexpressible in AlphaGeometry, **21.2% (55 theorems)** are partially expressible but not semantically equivalent. Only **21.9% (57 theorems)** theorems in LeanGeo can be completely translated in Alphageometry's pattern. On the other hand, **100%** of AlphaGeometry-expressible statements are expressible in LeanGeo.

Below are representative theorems from LeanGeo whose statements cannot be expressed in Alpha-Geometry due to limitations of its formal system.

**Example 1: Diameter is the longest chord.**

```
theorem diameter_longest :
  ∀ (a b c d o : Point) (C : Circle),
    (Diameter a b o C) ∧ (c.onCircle C) ∧ (d.onCircle C)
    → |(a-b)| ≥ |(c-d)| := by
```

AlphaGeometry does not support inequalities, so relations such as $|AB| \geq |CD|$ cannot be expressed at all.

**Example 2: Orthocenter of an acute triangle lies inside the triangle.**

```
theorem orthocentre_of_acuteTriangle_insideTriangle :
  ∀ (A B C H D E F : Point) (AB BC CA : Line),
    (formAcuteTriangle A B C AB BC CA) ∧
    (Orthocentre H A B C D E F AB BC CA)
    → InsideTriangle H A B C AB BC CA := by
```

AlphaGeometry cannot express "inside/outside" relations or "acute/obtuse" distinctions, making this theorem inexpressible.

Example 3: Existence of a circumcenter

```
theorem exists_circumcentre :
  ∀ (A B C : Point), Triangle A B C →
    ∃ (O : Point), Circumcentre O A B C := by
```

AlphaGeometry lacks existential quantifiers such as "there exists", so existence theorems cannot be stated.

**Example 4: Law of sines (radius form)**

```
theorem LawOfSines_radius :
  ∀ (A B C O: Point),
    Triangle A B C ∧ Circumcentre O A B C
    → |(B-C)| = 2 * Real.sin (\angle B:A:C) * |(A-O)| := by
```

AlphaGeometry does not include trigonometric functions and therefore cannot express any theorem involving sin, cos, or angle measure.

**Example 5: Cyclic quadrilateral angle relations.**

```
theorem cyclic_eq_angles' :
  ∀ (A B C D: Point) (AB : Line) (Ω : Circle),
    distinctPointsOnLine A B AB ∧
    C.sameSide D AB ∧
    A.onCircle Ω ∧ B.onCircle Ω ∧
    C.onCircle Ω ∧ D.onCircle Ω
    → \angle C:A:D = \angle C:B:D := by
```

AlphaGeometry uses unordered "full-angle" equality, which cannot distinguish positional relations or angle orientation, making this theorem not exactly expressible. In AlphaGeometry's framework, this statement is expressed as "cyclic A B P Q =¿ eqangle P A P B Q A Q B". This formulation does not account for changes in the relative positions of A, B, P, Q that may cause $\angle APB = \angle AQB$ or $\angle APB + \angle AQB = \pi$.

To further illustrate the differences between our formal system and that of AlphaGeometry in the shared subset of representation, we present the followiing two examples.

**Example 6:** Prove that the mid-segment of an isosceles trapezoid $ABCD$ is parallel to $AB$.

**LeanGeo proof:**

```
theorem trapezoid_midsegment_parallel_base :
  ∀ (A B C D E F: Point) (AB BC CD DA EF: Line),
  formQuadrilateral A B C D AB BC CD DA ∧
  (¬ AB.intersectsLine CD) ∧ distinctPointsOnLine E F EF ∧
  MidPoint B E C ∧ MidPoint A F D →
  (¬ EF.intersectsLine CD) := by
    euclid_intros
    euclid_apply line_from_points A E as AE
    euclid_apply intersection_lines CD AE as G
    have h1: |(A-E)| = |(E-G)| := by
      euclid_apply trapezoid_imp_similarTriangles_interior B A C G E AB
    CD
      euclid_apply similar_AA B A E C G E
      euclid_assert |(B-E)| = |(C-E)|
      euclid_apply congruentTriangles_ASA B E A C E G
      euclid_finish
    have h2: ¬ EF.intersectsLine CD := by
      euclid_apply triangleMidsegment_parallel_base A D G F E DA CD AE
      euclid_finish
    euclid_finish
```

**Alphageometry Proof:**

```
=========================
 * From theorem premises:
A B C D E F : Points
DC // AB [00]
A,E,C are collinear [01]
EA = EC [02]
F,B,D are collinear [03]
FB = FD [04]

 * Auxiliary Constructions:
: Points

 * Proof steps:
001. EA = EC [02] & FB = FD [04] ⇒ EA:EC = FB:FD [05]
002. CD // AB [00] & A,E,C are collinear [01] &
     F,B,D are collinear [03] & EA:EC = FB:FD [05]
     ⇒ EF // CD
=========================
```

The reason AlphaGeometry produces such a short proof is that its deductive database contains many relatively high-level secondary rules (as shown in step 002). These rules are treated as "axioms" inside AlphaGeometry. In contrast, within the LeanGeo framework, we do not freely introduce such axioms. Instead, all basic theorems must be proved from more primitive axioms and inference tools. For instance, in this problem we introduce an auxiliary intersection point of $CD$ and $AE$, and then complete the proof via congruence and similarity of triangles. As a consequence, our proof is longer but conceptually more instructive.

**Example 7: IMO 2000 P1**

```
Two circles G₁ and G₂ intersect at two points M and N. Let AB be the
    line tangent to these circles at A and B, respectively, so that M
    lies closer to AB than N. Let CD be the line parallel to AB and
    passing through the point M, with C on G₁ and D on G₂. Lines AC
    and BD meet at E; lines AN and CD meet at P; lines BN and CD
    meet at Q. Show that EP = EQ.
```

Listing 8: IMO 2000 Problem 1

**LeanGeo proof:**

```
import Mathlib
import SystemE
import LeanGeo
namespace LeanGeo
set_option maxHeartbeats 0
--To circle
--Two circles G₁ and G₂ intersect at two points M and N. Let AB be the
    line tangent to these circles at A and B, respectively, so that M
    lies closer to AB than N. Let CD be the line parallel to AB and
    passing through the point M, with C on G₁ and D on G₂. Lines AC
    and BD meet at E; lines AN and CD meet at P; lines BN and CD
    meet at Q. Show that EP = EQ.
theorem IMO_2000_P1 :
  ∀ (M N A B C D E P Q O1 O2 : Point) (G1 G2 : Circle) (AB CD AC BD AN
    BN : Line),
    CirclesIntersectAtTwoPoints G1 G2 M N ∧
    distinctPointsOnLine A B AB ∧
    TangentLineCircleAtPoint A O1 AB G1 ∧
    TangentLineCircleAtPoint B O2 AB G2 ∧
    ¬ AB.intersectsLine CD ∧
    distinctPointsOnLine M C CD ∧
    C.onCircle G1 ∧ C ≠ M ∧ C ≠ N ∧
```

```
D.onCircle G2 ∧ between C M D ∧
distinctPointsOnLine A C AC ∧
distinctPointsOnLine B D BD ∧
between E A C ∧ between E B D ∧
distinctPointsOnLine A N AN ∧
TwoLinesIntersectAtPoint AN CD P ∧
distinctPointsOnLine B N BN ∧
TwoLinesIntersectAtPoint BN CD Q →
|(E-P)| = |(E-Q)| := by
euclid_intros
euclid_apply line_from_points M N as MN
euclid_apply intersection_lines MN AB as T
have midP_ATB: MidPoint A T B := by
  have h1: |(T-A)| * |(T-A)| = |(T-M)| * |(T-N)| := by
    euclid_apply TangentSecantTheorem T A M N O1 G1 AB
    euclid_finish
  have h2: |(T-B)| * |(T-B)| = |(T-M)| * |(T-N)| := by
    euclid_apply TangentSecantTheorem T B M N O2 G2 AB
    euclid_finish
  have h3: |(T-A)| * |(T-A)| = |(T-B)| * |(T-B)| := by
    rw[h1,h2]
  euclid_assert |(T-A)| > 0
  euclid_assert |(T-B)| > 0
  have h4: |(T-A)| = |(T-B)| := by
    nlinarith
  euclid_finish
have midP_PMQ : MidPoint P M Q := by
  have h1 : |(M-Q)| = |(M-P)| := by
    have h4: |(T-A)| = |(T-B)| := by euclid_finish
    have h5: |(M-Q)| * |(T-A)| = |(M-P)| * |(T-B)| := by
      euclid_apply triangle_parallel_bases_eq_ratio N T A M P B Q AB
CD
      euclid_finish
    rw [h4] at h5
    have h6: |(T-B)| > 0 := by euclid_finish
    euclid_finish
  have h2: between P M Q := by
    euclid_finish
  euclid_finish
euclid_apply line_from_points E M as EM
have h_congr: CongruentTriangles A B E A B M := by
  have h1: ∠E:A:B = ∠M:A:B := by
    have h2: ∠E:A:B = ∠E:C:D := by
      euclid_apply parallel_imp_eq_alternateExteriorAngles B A D C E
AB CD AC
      euclid_finish
    have h3: ∠M:A:B = ∠M:C:A := by
      euclid_apply line_from_points A M as AM
      have h4: M.sameSide B AC := by
        euclid_finish
      euclid_apply AlternateSegmentTheorem A M C B O1 G1 AM CD AC AB
      euclid_finish
    euclid_finish
  have h5: ∠E:B:A = ∠M:B:A := by
    have h6: ∠E:B:A = ∠E:D:C := by
      euclid_apply parallel_imp_eq_alternateExteriorAngles A B C D E
AB CD BD
      euclid_finish
    have h7: ∠M:B:A = ∠M:D:B := by
      euclid_apply line_from_points B M as BM
      have h8: M.sameSide A BD := by
        euclid_finish
      euclid_apply AlternateSegmentTheorem B M D A O2 G2 BM CD BD AB
      euclid_finish
    euclid_finish
```

```
              euclid_apply congruentTriangles_ASA A B E A B M
              euclid_finish
          have perp_EM_CD: PerpLine EM CD := by
            have h1: PerpBisector E M AB := by
              euclid_apply perpBisector_if_eq_dist E M A B AB
              euclid_finish
            euclid_apply perpBisector_imp_perpLine E M EM AB
            euclid_apply perp_parallel_imp_perp AB EM CD
            euclid_finish
          have perpB: PerpBisector P Q EM := by
            euclid_apply (perpBisector_iff P Q EM).mpr
            euclid_finish
          euclid_finish
```

Listing 9: Proof of LeanGeo for IMO_2000_P1

**Alphageometry Proof:**

```
* Formal statement:
a b = segment a b; c = on_tline c a a b; d = on_tline d b b a; e =
    on_circle e c a, on_circle e d b; f = on_circle f c a, on_circle f d
    b; g = on_pline g e a b, on_circle g c a; h = on_pline h e a b,
    on_circle h d b; i = on_line i a g, on_line i b h; j = on_line j a
    f, on_line j g h; k = on_line k b f, on_line k g h ? cong i j i k
==========================
 * From theorem premises:
A B C D E F G H I J K : Points
AC ⊥ AB [00]
BA ⊥ DB [01]
DE = DB [02]
CE = CA [03]
DF = DB [04]
CF = CA [05]
∠FAE = ∠FAE [06]
GE // AB [07]
CG = CA [08]
∠GAF = ∠GAF [09]
HE // AB [10]
DH = DB [11]
∠FBH = ∠FBH [12]
I,G,A are collinear [13]
I,B,H are collinear [14]
J,F,A are collinear [15]
J,G,H are collinear [16]
BF:BK = BF:BK [17]
G,K,H are collinear [18]
B,F,K are collinear [19]

 * Auxiliary Constructions:
: Points

 * Proof steps:
001. EG // AB [07] & EH // AB [10] ⇒  EH // EG [20]
002. EH // EG [20] ⇒  E,G,H are collinear [21]
003. DH = DB [11] & DF = DB [04] ⇒  D is the circumcenter of \Delta BHF
    [22]
004. D is the circumcenter of \Delta BHF [22] & DB ⊥ BA [01] ⇒  ∠ABH = ∠
    BFH [23]
005. D is the circumcenter of \Delta BHF [22] & DB ⊥ BA [01] ⇒  ∠ABF = ∠
    BHF [24]
006. E,G,H are collinear [21] & G,K,H are collinear [18] & ∠BFH = ∠ABH
    [23] & AB // EG [07] ⇒  ∠BFH = ∠KHB [25]
```

007. E,G,H are collinear [21] & G,K,H are collinear [18] & B,F,K are
     collinear [19] & ∠BHF = ∠ABF [24] & AB // EG [07] ⇒  ∠BHF = ∠HKB
     [26]
008. ∠BFH = ∠KHB [25] & ∠BHF = ∠HKB [26] (Similar Triangles)⇒  BF:BH =
     BH:BK [27]
009. DF = DB [04] & DH = DB [11] & DE = DB [02] ⇒  E,B,F,H are
     concyclic [28]
010. DF = DB [04] & DE = DB [02] ⇒  D is the circumcenter of \Delta BFE
     [29]
011. D is the circumcenter of \Delta BFE [29] & DB ⊥ BA [01] ⇒  ∠EBA = ∠
     EFB [30]
012. E,G,H are collinear [21] & ∠EFB = ∠EBA [30] & AB // EG [07] ⇒  ∠
     EFB = ∠BEH [31]
013. E,B,F,H are concyclic [28] & ∠EFB = ∠BEH [31] ⇒  EB = BH [32]
014. CE = CA [03] & CG = CA [08] ⇒  C is the circumcenter of \Delta AEG
     [33]
015. C is the circumcenter of \Delta AEG [33] & AC ⊥ AB [00] ⇒  ∠BAE = ∠
     AGE [34]
016. I,G,A are collinear [13] & ∠BAE = ∠AGE [34] & EG // AB [07] ⇒  ∠
     IAB = ∠BAE [35]
017. DH = DB [11] & DE = DB [02] ⇒  D is the circumcenter of \Delta BHE
     [36]
018. D is the circumcenter of \Delta BHE [36] & DB ⊥ BA [01] ⇒  ∠ABH = ∠
     BEH [37]
019. I,B,H are collinear [14] & ∠ABH = ∠BEH [37] & EH // AB [10] ⇒  ∠
     ABE = ∠IBA [38]
020. ∠IAB = ∠BAE [35] & ∠ABE = ∠IBA [38] (Similar Triangles)⇒  BI = BE
     [39]
021. ∠IAB = ∠BAE [35] & ∠ABE = ∠IBA [38] (Similar Triangles)⇒  AI = AE
     [40]
022. BF:BH = BH:BK [27] & EB = BH [32] & BI = BE [39] ⇒  IB:BF = BK:IB
     [41]
023. B,F,K are collinear [19] & I,B,H are collinear [14] & ∠FBH = ∠FBH
     [12] ⇒  ∠KBI = ∠FBI [42]
024. IB:BF = BK:IB [41] & ∠KBI = ∠FBI [42] (Similar Triangles)⇒  BK:IK =
     IB:IF [43]
025. E,B,F,H are concyclic [28] ⇒  ∠FEH = ∠FBH [44]
026. CF = CA [05] & CG = CA [08] & CE = CA [03] ⇒  E,G,F,A are
     concyclic [45]
027. E,G,F,A are concyclic [45] ⇒  ∠GEF = ∠GAF [46]
028. I,G,A are collinear [13] & I,B,H are collinear [14] & ∠FEH = ∠FBH
     [44] & EH // AB [10] & ∠GEF = ∠GAF [46] & EG // AB [07] ⇒  ∠IAF = ∠
     IBF [47]
029. ∠IAF = ∠IBF [47] ⇒  I,B,F,A are concyclic [48]
030. I,B,F,A are concyclic [48] ⇒  ∠IBA = ∠IFA [49]
031. I,B,F,A are concyclic [48] ⇒  ∠IFB = ∠IAB [50]
032. E,G,H are collinear [21] & G,K,H are collinear [18] & J,F,A are
     collinear [15] & ∠IBA = ∠IFA [49] & I,B,H are collinear [14] & ∠ABH =
     ∠BEH [37] & EH // AB [10] & AB // EG [07] ⇒  ∠BEK = ∠JFI [51]
033. CE = CA [03] & CF = CA [05] ⇒  C is the circumcenter of \Delta AEF
     [52]
034. C is the circumcenter of \Delta AEF [52] & AC ⊥ AB [00] ⇒  ∠BAE = ∠
     AFE [53]
035. J,G,H are collinear [16] & E,G,H are collinear [21] & ∠BAE = ∠AFE
     [53] & AB // EG [07] ⇒  ∠JEA = ∠AFE [54]
036. J,F,A are collinear [15] & ∠FAE = ∠FAE [06] ⇒  ∠JAE = ∠FAE [55]
037. ∠JEA = ∠AFE [54] & ∠JAE = ∠FAE [55] (Similar Triangles)⇒  JA:EA =
     EA:FA [56]
038. EA:FA = JA:EA [56] & IA = EA [40] ⇒  IA:FA = JA:IA [57]
039. I,G,A are collinear [13] & J,F,A are collinear [15] & ∠GAF = ∠GAF
     [09] ⇒  ∠IAF = ∠IAJ [58]
040. IA:FA = JA:IA [57] & ∠IAF = ∠IAJ [58] (Similar Triangles)⇒  ∠AIF =
     ∠IJA [59]
041. B,F,K are collinear [19] & E,G,H are collinear [21] & G,K,H are
     collinear [18] & J,F,A are collinear [15] & ∠AIF = ∠IJA [59] & I,G,A

```
        are collinear [13] & ∠IFB = ∠IAB [50] & AB // EG [07] ⇒  ∠BKE = ∠
        FJI [60]
042. ∠BEK = ∠JFI [51] & ∠BKE = ∠FJI [60] (Similar Triangles)⇒  BE:IF =
        BK:IJ [61]
043. BK:IK = IB:IF [43] & BE:IF = BK:IJ [61] & BI = BE [39] ⇒  BK:JI =
        BK:IK [62]
044. BF:BK = BF:BK [17] & BK:JI = BK:IK [62] ⇒  JI = IK
        =========================
```

Listing 10: Proof of AlphaGeometry for IMO 2000 Problem 1

AlphaGeometry presents the proof as a flat, linear sequence of 44 atomic deductions. While logically sound, this format obscures the underlying geometric narrative. It reads as a symbolic log where high-level concepts, without explicitly grouping these steps into a coherent subgoal.

In contrast, the LeanGeo proof is structured more hierarchically, perfectly reflecting the problem's intrinsic geometric structure. The proof is organized into clear, self-contained logical blocks, such as proving 'midP_ATB' (T is the midpoint of AB) or 'perp_EM_CD'. Each block is achieved by invoking powerful theorems in LeanGeo library like 'TangentSecantTheorem' and 'AlternateSegmentTheorem' — mirroring the exact language a mathematicia would use. Consequently, the Lean-Geo proof is not only verifiable but also intelligible, bridging the gap between a machine-generated proof trace and a human-authored mathematical argument. It demonstrates a system that reasons in a manner remarkably close to natural geometric intuition.

### E.2 VERIFIABILITY AND SOUNDNESS

A fundamental requirement for any formal deductive system is **soundness**: every statement that can be derived within the system must be logically valid under the intended semantics. In other words, a proof system is sound if it never proves anything false.

One important limitation of AlphaGeometry is that it can only **generate** correct proofs, but cannot **verify** them. Each proof generated by AlphaGeometry implicitly corresponds to a specific geometric figure, and the deductions are valid only within that configuration. For other admissible figures satisfying the same hypotheses, the conclusion may fail.

**Example 8: The internal angle bisector and the external angle bisector are perpendicular.**

AlphaGeometry's Proof:

```
Input:
b c d = triangle b c d; a = on\_line a b d; e = angle\_bisector e b a c;
    f = angle\_bisector f c a d ? perp e a a f
=========================
 * From theorem premises:
   B C D A E F : Points
   D,A,B are collinear [00]
   \angle BAE = \angle EAC [01]
   \angle CAF = \angle FAD [02]
* Proof steps:
1. \angle CAF = \angle FAD [02] & D,A,B are collinear [00] ⇒  \angle
    CAF = \angle FAB [03]
2. \angle BAE = \angle EAC [01] & \angle CAF = \angle FAB [03] (Angle
    chase) ⇒  AE \perp AF
=========================
```

However, if claim that A,E,F are collinear, AlphaGeometry produces a completely contradictory conclusion under exactly the same assumptions.

```
Input:
b c d = triangle b c d; a = on\_line a b d; e = angle\_bisector e b a c;
    f = angle\_bisector f c a d ? coll e a f
=========================
 * From theorem premises:
```

```
B C D A E F : Points
A,D,B are collinear [00]
\angle BAE = \angle EAC [01]
\angle CAF = \angle FAD [02]
 * Auxiliary Constructions:
: Points

 * Proof steps:
001. \angle CAF = \angle FAD [02] & A,D,B are collinear [00] ⇒  \angle
     CAF = \angle FAB [03]
002. \angle BAE = \angle EAC [01] & \angle CAF = \angle FAB [03] (Angle
     chase) ⇒  AE // AF [04]
003. AE // AF [04] ⇒  E,F,A are collinear
     ==========================
```

The core issue is that many of AlphaGeometry's built-in inference rules are not purely syntactic logical consequences of axioms; instead, they depend on properties of the internal geometric diagram. Since this diagram-based reasoning is not exposed or verified independently of the figure, ambiguous or under-specified statements may lead to incorrect deductions.

LeanGeo, however, is graph-free and handles positional relations with full logical rigor. This inevitably makes its proofs more complex, but we believe it more faithfully reflects the intrinsic nature of geometric reasoning.

Overall, AlphaGeometry is a *task-specialized solving system* tailored for IMO-style geometry problems: it is extremely powerful in problem solving, but this comes at the cost of sacrificing internal axiomatic rigor and omitting several components we believe are equally essential for geometry learners and researchers—such as geometric inequalities, trigonometric reasoning, and positional or incidence relations. Its simplified formal system accelerates search and inference but loses part of the rigor and human interpretability. In contrast, our system aims to be more complete, rigorous, and structurally expressive, though this naturally results in more intricate and elaborate reasoning processes.

## F  PROMPT FOR EVALUATION

```
    You are an expert of Lean 4. Now You are using a new Lean 4 system
    called LeanEuclid. The following is how you prove your theorem.
--- Proof DSL ---
Your proof must be a tactic proof in the LeanEuclid proof DSL. This DSL
    is built from
    the following tactics (arguments shown in angle-brackets <> ):
* TACTIC: euclid_intros *
 Introduces universally quantified variables and premises of the current
    goal into the proof context. No names required.
* TACTIC: euclid_apply <rule> <args> *
where <rule> is either a construction rule, inference rule, or other
    theorem.
Given a rule <rule> with type of the form ∀ (<args> : Types) ... P -> Q,
    this tactic
     instantiates <rule> with <args>, and attempts to prove premise P
    from the local proof
      context using an SMT solver. If successful, propsition Q is added
    to the proof
      context.
usage examples :
  euclid_apply PythagoreanTheorem_point a b c : SMT solver will try to
    search whether the premise of theorem "PythagoreanTheorem_point"
    i.e.(Triangle a b c) ∧ (∠ b:a:c : ℝ) are satisfied, if not, the
    proof will fail. If all premises are found, then the conclusion of
    this theorem will be added to the solving context, i.e. |(b−c)| * |
    (b−c)| = |(b−a)| * |(b−a)| + |(a−c)| * |(a−c)|.
```

```
* TACTIC: euclid_apply <rule> <args> as X *
Given a rule <rule> with type of the form ∀ (<args> : Types) ... P -> ∃ x
    . Q(x), this
    tactic instantiates <rule> with <args>, and attempts to prove
    premise P from the local
    proof context using an SMT solver. If successful, object x and
    premise Q(x) are added
    to the proof context.
usage examples:
  euclid_apply line_from_points p1 p2 as M this tactic will first check
    whether p1 and p2 are different. If they are, then a new line M is
    added to the proof context and new condition, p1.onLine M and
    p2.onLine M will be added to the condition.

NOTE: You can only use 'euclid_apply  <rule> <args> as <X>' if the rule
    produces an
    existential. You should not name any propsotions introduced using
    'euclid_apply' e,g,
    'euclid_apply <rule> <args> as H1'.
NOTE: It is very important that *all* non-propositional (i.e.,
    universally quantified)
    arguments are provided to the rule when invoking 'euclid_apply'.
*TACTIC: euclid_finish *
    Attempts to resolve the proof goal using the current proof context
    using an SMT solver.

* euclid_assert <P> *
    Attempts to prove proposition <P> from the current proof context
    using an SMT solver.
        Equivalent to "have : <P> := by euclid_finish"

If you are proving an existentially quantified proposition, you can use
    the standard Lean tactic ' use <X>' to provide the witness <X> for
    the quantifier. DO NOT use the tactic 'use' if you are not proving
    an existentially quantified proposition.

Here is several additional tips with examples:

1. You can use standard Lean tactics such as <by_cases>, <cases>,
    <split_ands> and <constructor> <by_contra> to structure your proof.
    Specifically, you are encouraged to use "have hX: P := by" to divide
    the whole problems to small proposition. However, you should not use
    imperative Lean tactics, such as 'rw' or 'simp'. You should only use
    the above declarative tactics.

2. You should be careful to check the degenerate case and special cases.
    For example, sometimes you want to get the intersection of two
    lines. You may use"euclid_apply intersection_lines L1 L2 as O" but
    before that you should guarantee that the SMT can deduce that L1 and
    L2 intersects.

3. You must ensure that every step in your proof is rigorous, not only
    in natural language, but in LeanEuclid. For example, in the
    following proof,
<error_example1>
theorem altitude_hypotenuse_similar:
  ∀ (A B C D: Point) (BC : Line),
    RightTriangle A B C ∧
    distinctPointsOnLine B C BC ∧
    foot A D BC
    → SimilarTriangles D B A A B C := by
    euclid_intros
    have h_tri_DBA : Triangle D B A := by
      euclid_finish"
    ...
```

```
<correction1>
Here if you want to claim triangle D B A, you must either prove that D
    is not equal to B and A, or claim it in your premise (like adding
    between A B D). Although in natural language it is trivial, but in
    this formal language you must PROVE it! In this example, instead,
    your method to prove h_tri_DBA should be:
    have h_tri_DBA : Triangle D B A := by
      have h4: between C D B := by
        have h5: ∠A:B:C < ∟:= by
           euclid_apply triangle_angles_sum A B C
           euclid_finish
        have h6: ∠A:C:B < ∟:= by
           euclid_apply triangle_angles_sum A B C
           euclid_finish
        euclid_apply acuteTriangle_foot_between A B C D BC
        euclid_finish
      euclid_finish.
NOTE: Using recursive "have"s to split the goal and make the proof neat.

Another example is:
<error_example2>
theorem apollonius_isoceles :
  ∀ (A B C D : Point) (BC : Line),
    IsoTriangle A B C ∧
    distinctPointsOnLine B C BC ∧
    Coll B D C ∧
    between B D C
    → |(A-B)| * |(A-B)| - |(A-D)| * |(A-D)| = |(B-D)| * |(C-D)| := by
  euclid_intros
  have h_A_not_on_BC : ¬(A.onLine BC) := by
    euclid_finish
  euclid_apply exists_foot A BC as H
  have h_midpoint_H : MidPoint B H C := by
    euclid_apply isoTriangle_three_lines_concidence_foot A B C H BC
    euclid_finish
  have h_tri_AHD : Triangle H A D := by
    euclid_finish

<correction2>
  Here h_tri_AHD is wrong. Since you cannot assume triangle H A D,
    because H may coincide with D. Instead your response shold be:
    by_cases H = D
    · ...
    · have h_tri_AHD : triangle H A D := by
        -- H, D are on line BC, while A is not. So H, A, D are not
    collinear.
        euclid_finish
      ...

<error_example3>
theorem Numina_Geometry_1110 :
  ∀ (A B C H M K : Point) (AC: Line),
    (triangle A B C) ∧
    (between A H C) ∧
    (foot B H AC) ∧
    (distinctPointsOnLine A C AC) ∧
    (midpoint B M C) ∧
    (midpoint A K B)
    →
    (∠ K:H:M = ∠ A:B:C)
    euclid_intros
    have h_tri_KHM: triangle K H M := by euclid_finish
    ...
```

```
3. "euclid_assert" make very few progress in the proof. Try to use less
   "euclid_assert X", but use more "have h: X := by ...".

4. When using the "*" symbol for multiplication, please ensure there is
   a space on both sides of the "*" symbol. For example, the correct
   expression should be "|(A-M)| * |(B-M)|" instead of "|(A-M)|*|(B-M)|"

5. Sometimes when chasing angles, especially using "coll_angles_eq" and
   "coll_supp_angles" you are encouraged to use "line_from points" to
   construct the between-line, for example, in the following theorem,
<error_example>
theorem median_is_half_side_implies_right_triangle:
  ∀ (A B C M : Point),
    Triangle A B C ∧
    MidPoint B M C ∧
    |(A-M)| = |(B-M)|
    → ∠ B:A:C = ∟ := by
    have h_sum_BAC : ∠B:A:M + ∠M:A:C = ∠B:A:C := by
      euclid_apply coll_supp_angles A B M C
      euclid_finish

<correction>
In the example, "euclid_apply coll_supp_angles A B M C" will fail
    because the SMT cannot deduce A,B,M form a triangle. So how to prove
    this? actually you should add a line "euclid_apply line_from_points
    B C as BC" in your proof. Remember SMT cannot construct. So you
    should tell SMT there is a line BC, and SMT will automatically
    deduce A B M are not collinear. So your proof should be
theorem median_is_half_side_implies_right_triangle:
  ∀ (A B C M : Point),
    Triangle A B C ∧
    MidPoint B M C ∧
    |(A-M)| = |(B-M)|
    → ∠ B:A:C = ∟ := by
    have h_sum_BAC : ∠B:A:M + ∠M:A:C = ∠B:A:C := by
      euclid_apply line_from_points B C as BC.
      euclid_apply coll_supp_angles A B M C
      euclid_finish

6. Take care of the order of parameter. For example, if you want to
   express "Right Triangle ABC with right angle ABC", you should use
   "RightTriangle B A C" (First parameter is rightangle) instead of
   "rightTriangle A B C". When apply lemma or writing formal statement,
   always check whether the order is align with definition. Also you
   should check the number of parameters. For example, "Coll" only
   contains three parameters. So donn't use "Coll A B C D" to represent
   A,B,C,D are collinear. Instead, use "Coll A B C ∧ Coll B C D"

7. At the beginning of your proof, you should firstly using
   "euclid_apply line_from_points X Y as XY" To obtain all the the line
   you needed in the problem, if the problem does not give these lines.
   This step is benificial to the later SMT steps.

8. When using "euclid_apply", do not add additional condition to it, for
   example, do not use "euclid_apply coll_supp_angles A E C B
   h_between_AEC hA". Instead, use "euclid_apply coll_supp_angles A E C
   B". SMT will automatically search whether the absent condition is
   satisfied.
--- End of Proof DSL ---

Your proofs can make use of the following abbreviation of geometry
   structure:
--- Begin of Abbreviation ---
```

```
1674  /-Relations-/
1675
1676  abbrev Coll (A B C : Point) : Prop :=
1677  between A B C ∨ between B C A ∨ between C A B ∨ A = B ∨ A = C ∨ B = C
1678
1679  abbrev Triangle (A B C : Point) : Prop :=
1680  ¬ (Coll A B C)
1681  ...
1682  ...
1683
1684  abbrev RadicalAxis (Ω₁ Ω₂ : Circle) (L : Line) : Prop :=
1685  ∀ (A : Point), A.onLine L → Pow(A, Ω₁) = Pow(A, Ω₂)
1686  --- End of Abbreviation ---
1687
      Also, I'll provide you the construction rules where you can construct
1688      lines, points and circles by these rules using "euclid_apply
1689      <theorem> as ...". Notice that these rules are not included in SMT.
1690      So you should construct lines, points in your proof by yourself.
1691
      --- Begin of Construction Rules ---
1692
1693  axiom intersection_lines : ∀ (L M : Line), L.intersectsLine M →
1694    ∃ a : Point, (a.onLine L) ∧ (a.onLine M)
1695  utsideCircle α
1696
1697  ...
      ...
1698
1699  axiom exists_distinct_point_outside_circle :
1700    ∀ (α : Circle) (b : Point),  ∃ a : Point, a ≠ b ∧ a.outsideCircle α
1701
      --- End of Construction Rules ---
1702  Also, I'll provide you the theorem libarary. Use "euclid_apply" to use
1703      these theorems in theorem library.
1704  --- Begin of Theorem Library ---
1705
1706  axiom triangle_area_foot :∀ (a b c d: Point) (BC: Line),b.onLine BC ∧
1707      c.onLine BC ∧ (Triangle a b c) ∧ Foot a d BC → (△a:b:c).area = |
1708      (a-d)| * |(b-c)|/2
1709
1710  ...
      ...
1711
1712  theorem trapezoid_midsegment_parallel_base : ∀ (A B C D E F: Point) (AB
1713      BC CD DA EF: Line), formQuadrilateral A B C D AB BC CD DA ∧ (¬
1714      AB.intersectsLine CD) ∧ distinctPointsOnLine E F EF ∧ MidPoint B E C
      ∧ MidPoint A F D →  (¬ EF.intersectsLine CD) := by
1715  --- End of Theorem Library ---
1716  All theorems in library are proved and you can apply them directedly.
1717      The following are few-shot example proof of the most commonly used
1718      theorems in library.
1719  --- Few-shot Examples ---
      Input1:
1720  import Mathlib
1721  import SystemE
1722  import LeanGeo
1723  namespace LeanGeo
1724  theorem InscribedAngleTheorem_sameSide :
1725    ∀ (A B C O : Point) (AB: Line) (Ω : Circle), Triangle A B C ∧
1726      distinctPointsOnLine A B AB ∧ (O.sameSide C AB) ∧ (A.onCircle Ω) ∧
1727      (B.onCircle Ω) ∧ (C.onCircle Ω) ∧ (O.isCentre Ω)
        → ∠ A:O:B = ∠ A:C:B + ∠ A:C:B := by
```

```
Output1:
import Mathlib
import SystemE
import LeanGeo
namespace LeanGeo

...
...

Output5:
import Mathlib
import SystemE
import LeanGeo
namespace LeanGeo

theorem cyclic_supp_angles : ∀ (A B C D: Point) (AB:Line) (Ω : Circle),
    distinctPointsOnLine A B AB ∧ DistinctFourPoints A B C D ∧
    A.onCircle Ω ∧ B.onCircle Ω ∧  C.onCircle Ω ∧ D.onCircle Ω ∧
    C.opposingSides D AB → ∠B:C:A + ∠B:D:A = ∟ + ∟ := by
  euclid_intros
  euclid_apply exists_centre Ω as O
  by_cases O.sameSide C AB
  · euclid_assert O.opposingSides D AB
    euclid_apply InscribedAngleTheorem_sameSide A B C O AB Ω
    euclid_apply InscribedAngleTheorem_opposingSides A B D O AB Ω
    euclid_finish
  · by_cases O.onLine AB
    · euclid_apply ThalesTheorem A B C O Ω
      euclid_apply ThalesTheorem A B D O Ω
      euclid_finish
    · euclid_apply InscribedAngleTheorem_sameSide A B D O AB Ω
      euclid_apply InscribedAngleTheorem_opposingSides A B C O AB Ω
      euclid_finish
-- End of Few-shot Examples ---

IMPORTANT: Your response should be started with
"import Mathlib
import SystemE
import LeanGeo
namespace LeanGeo

theorem ..." You should restate the theorem that you want to prove in
    formal language, give a complete proof of the theorem.
Now, please prove the following theorem:
<formal statement>
```

Listing 11: Prompt for LLMs in Evaluation

# G   LLM ACKNOWLEDGMENTS

The authors acknowledge the use of an AI-powered language tool (e.g., ChatGPT, GPT-4) for enhancing the readability of this paper. We wish to clarify that all core ideas, research frameworks, and expressed opinions are original to the authors. Furthermore, all experimental results and data reported are authentic and based on real-world tests conducted by our team. The authors assume full responsibility for the content and integrity of this work.

