# OpenReview forum: "LeanGeo: Formalizing Competitional Geometry problems in Lean"
_ICLR.cc/2026/Conference — Submitted to ICLR 2026_

### Official Review · Reviewer_iGd7 · 2025-10-27

**Soundness:** 3
**Presentation:** 3
**Contribution:** 3
**Rating:** 4
**Confidence:** 2

**Summary:**

A unified framework in the Lean 4 theorem prover for formalizing competition-level geometry problems. It features a comprehensive library of 260 theorems and LeanGeo-Bench, a benchmark of 122 problems including 43 IMO geometry tasks, to evaluate large language models' reasoning capabilities, highlighting current limitations and the need for further advancements.

**Strengths:**

LeanGeo provides a novel framework that integrates competition-level geometry into Lean 4, complemented by LeanGeo-Bench, offering a valuable resource for AI reasoning research. The seamless integration with Mathlib enables LeanGeo to leverage algebraic and inequality tools, enhancing its applicability to complex, interdisciplinary mathematical problems.

**Weaknesses:**

1. LeanGeo heavily relies on existing frameworks such as LeanEuclid (Murphy et al., 2024) and SystemE (Avigad et al., 2009), with its primary contributions being an expanded theorem library and 52 new abbreviations (syntactic sugar). This incremental approach lacks substantial novelty. Compared to AlphaGeometry (Trinh et al., 2024), which introduces innovative neurosymbolic search, LeanGeo’s “human-like” proofs show limited differentiation. Table 1 highlights qualitative differences but fails to provide quantitative metrics, such as proof length or success rates on shared problems, to substantiate its advantages.

2. The statement ‘We present the first framework in the Lean theorem prover capable of expressing and reasoning about competition-level geometry problems in a human-like manner’ (Page 2, Lines 100-104) may require further clarification. Myers (2024) has demonstrated progress in formalizing planar geometry in Lean, including a solution to a 2019 IMO problem, as noted in community discussions (e.g., ‘Lean in 2024’ blog). While this suggests prior efforts in handling competition-level geometry, the scope and capabilities of Myers’ work compared to LeanGeo remain unclear.

3. The RL experiments in Section 5 are vague and lack depth. No ablation studies (e.g., assessing the impact of theorem library size or prompt strategies), hyperparameter details, or error analyses (e.g., identifying failed problems and their reasons) are provided. Additionally, the absence of comparisons with state-of-the-art methods, such as DeepSeek-Prover or Seed-Prover, undermines the validity of the reported “promising initial results.

4. LeanGeo-Bench, comprising 122 problems including 43 IMO geometry problems, appears limited in scale compared to existing benchmarks like MATP-BENCH (1,056 problems). This raises concerns about whether the dataset is sufficiently representative of the diversity and complexity of competition-level geometry problems.

**Questions:**

None

---

> ### Author Response · Authors · 2025-11-20
> **Response to Reviewer iGd7 Part 1/2**
>
> **W1. Comparing LeanGeo with AlphaGeometry**
>
> *"LeanGeo heavily relies on existing frameworks...Table 1 highlights qualitative differences but fails to provide quantitative metrics, such as proof length or success rates on shared problems, to substantiate its advantages."
>
> We acknowledge that our work builds upon and extends prior research. While this may reduce standalone novelty to some extent, it ensures rigor and continuity—both essential for developing a coherent and sound system for formal geometry.
>
> Regarding AlphaGeometry, its neurosymbolic axiomatic system and search methodology heavily rely on the Deductive Database of Chou et al. (2000), with its main innovation being the use of LLMs to propose auxiliary constructions. However, as we discuss in lines 40–48 and 155–164 of our paper, the AlphaGeometry system also suffers from several structural limitations. **We organize these limitations and include a detailed comparison in the revised Appendix E, where we additionally provide side-by-side examples contrasting our theorem-library proofs with those produced by AlphaGeometry.**
>
> Overall, AlphaGeometry is a *task-specialized solving system* tailored for IMO-style geometry problems: it is extremely powerful in problem solving, but this comes at the cost of sacrificing internal axiomatic rigor and omitting several components we believe are equally essential for geometry learners and researchers—such as geometric inequalities, trigonometric reasoning, and positional or incidence relations. Its simplified formal system accelerates search and inference but loses part of the rigor and human interpretability. In contrast, our system aims to be more complete, rigorous, and structurally expressive, though this naturally results in more intricate and elaborate reasoning processes.
>
> **W2. Clarifying LeanGeo's Relation to Myers (2024)**
>
> *“The claim of being the first framework in Lean for competition-level geometry needs clarification… Myers (2024) solved one IMO problem.”*
>
>  **Response:**
>
> We are also aware of Myers’ impressive and creative work. However, our contributions differ from his in several important aspects:
>
> 1. **Scope and Purpose.**
>
>    Myers’ work does not build a general *framework* for Euclidean geometry in Mathlib; rather, it only provides **one** isolated formalization of a single problem. In contrast, our work develops a complete and reusable **geometric theorem-proving framework**, together with a systematically constructed theorem library.
>
> 2. **Readability and Abstraction Level.**
>
>    Myers’ proof is highly formal and tightly coupled to the internal structures of Mathlib. For readers who are not already familiar with Mathlib’s geometry library, the proof is difficult to follow.
>    In comparison, **LeanGeo adopts a much more user-friendly and human-readable language**, which makes the proofs easier to read and understand.
>
> 3. **Proof Length and Expression Efficiency.**
>
>    Myers’ formalization requires roughly **600 lines**. In contrast, within LeanGeo’s framework, both of our IMO-level formal proofs require **around 100 lines**, reflecting the expressive efficiency provided by our framework and theorem library.
>
> **W3:Lack of ablation experiments in RL training**
>
> *"The RL experiments in Section 5 are vague and lack depth."*
>
> **Response:**
>
> We acknowledge that our RL exploration is indeed preliminary and does not include ablation studies. During training, we adjusted several hyperparameters (e.g., the ratio of negative gradients, reward shaping strategies) according to practical observations, but due to limited computational resources, we were unable to conduct systematic comparisons across hyperparameter settings. We also feel that the tuning process is engineering-heavy and orthogonal to the core contributions of the paper, which is why these details were not included in the main text.
>
> Regarding comparisons with existing provers, we tested several proof models that were specifically trained on miniF2F, such as DeepSeek-Prover and Kimina-Prover on LeanGeo-bench. Without additional SFT, their accuracy on LeanGeo is essentially zero because they do not support injecting the extensive domain knowledge required for our benchmark—this limitation is in fact one of the motivations for exploring our RL-based approach. As for the RL training method suggested by the reviewer, we note that both DeepSeek-Prover and Seed-Prover do not have fully open-sourced training pipelines, and their parameter scales far exceed our available computational capacity. Moreover, Seed-Prover itself is not open-source, preventing us from conducting any evaluation on it. For these reasons, a direct empirical comparison is unfortunately infeasible.

---

> > ### Author Response · Authors · 2025-11-20
> > **Response to Reviewer iGd7 Part 2/2**
> >
> > **W4: Scale of LeanGeo-Bench**
> >
> >  *“LeanGeo-Bench, comprising 122 problems including 43 IMO geometry problems, appears limited in scale compared to existing benchmarks like MATP-BENCH (1,056 problems). This raises concerns about whether the dataset is sufficiently representative of the diversity and complexity of competition-level geometry problems.”*
> >
> > **Response:**
> >
> >  We appreciate the concern regarding the number of problems in LeanGeo-Bench. During preparation of the RL training data, we experimented with autoformalizing **all geometric problems in Numina-math**—amounting to roughly 10,000 problems using the LeanGeo paradigm—and additionally used Gemini to generate 984 problems in LeanGeo format with correct proofs. These problems could potentially be incorporated into the benchmark. With additional manual annotation and filtering, the benchmark could be scaled to the order of 1,000 problems.
> >
> > However, we deliberately maintained a relatively small benchmark of 122 problems for several reasons:
> >
> > 1. **Evaluation cost:** Each problem is accompanied by up to 32K tokens of LeanGeo theorem-library context for LLM evaluation. Expanding the number of problems substantially would make evaluation prohibitively expensive.
> >
> > 2. **Efficiency for RL training:** In our RL experiments, the benchmark is tested frequently. Given the scarcity of Lean-geometry training data, a smaller benchmark ensures practical training efficiency.
> >
> > 3. **Representativeness:** We carefully curated the 122 problems to cover most typical competition-level geometric proofs—from middle-school to IMO level—across all main problem types (triangles, quadrilaterals, circles) and difficulty levels. Geometric reasoning is a relatively narrow and self-contained domain, and we believe this selection sufficiently reflects the diversity of competition geometry.
> >
> > Importantly, a large number of formalized problems alone does not guarantee meaningful evaluation. For instance, MATP-Bench contains many problems but does not provide the corresponding theorem-library tools; without access to fundamental geometric concepts like congruence or similarity, even middle-school level problems are essentially unsolvable for a prover. Currently, most problems in MATP-Bench lack the necessary tools for LLMs to solve them. In contrast, LeanGeo-Bench focuses on a curated subset of problems for which both the tools and formal framework are provided, ensuring that competition-level geometry problems can be meaningfully solved. This approach justifies the benchmark’s compact yet high-quality design. **Furthermore, our revised paper includes a detailed comparison between LeanGeo-Bench and existing Lean and geometry benchmarks to contextualize its design and contributions.**
> >
> > We again thank the reviewer for the valuable feedback. The comments have significantly improved the clarity and positioning of the paper. Let us know if additional clarifications are needed.

---

### Official Review · Reviewer_JjNZ · 2025-10-31

**Soundness:** 2
**Presentation:** 3
**Contribution:** 2
**Rating:** 4
**Confidence:** 2

**Summary:**

The paper introduces LeanGeo, a Lean-4–native framework and benchmark for competition-level Euclidean geometry. It offers a high-level theorem/tactic library, SMT integration, and a curated benchmark intended to mix geometry with general math (e.g., trigonometry, inequalities) via Mathlib.

**Strengths:**

- The paper presents a clear structure and provides readable examples; it is well-written, although it has minor wording issues.
- The system design is plausible and well-motivated.
- It is a valuable step toward Lean-native, competition-level geometry with cross-domain reasoning.

**Weaknesses:**

- The experiments are thin and largely non-diagnostic. There are no core ablations—no SMT off/partial settings, no scaling with library size or lemma granularity, no comparisons of tactic schedules, prompts, or decoding—so it’s unclear why models fail.
- It’s also unclear how this benchmark compares to others. There’s no side-by-side test on the same problems against existing Lean or geometry benchmarks, so claims about being better or different are mostly qualitative.
- Scalability and complexity are uncharacterized (no curves vs. points/constraints/branching), and the RL component is preliminary without stronger curricula or knowledge-tracing.

Overall, the experiments are too shallow to support firm claims. Adding these basic comparisons and reports would make the paper much stronger.

**Questions:**

- How does performance change with SMT off or with restricted solver capabilities?
- What are scaling curves for success/time vs. theorem-library size and lemma granularity?
- Can you provide aligned comparisons with existing Lean/geometry benchmarks on a shared subset?

---

> ### Author Response · Authors · 2025-11-20
> **Response to Reviewer JjNZ Part 1/2**
>
> We sincerely thank the reviewer for the thoughtful and constructive comments. We address each weakness and question below, and we have revised the paper accordingly.
>
> **Q1:Benchmark Comparison**
>
> *"There’s no side-by-side test on the same problems against existing Lean or geometry benchmarks."*
>
> *Can you provide aligned comparisons with existing Lean/geometry benchmarks on a shared subset?"*
>
> **Response**
> Thank you for this thoughtful suggestion. We fully agree that aligned comparisons are valuable. Here is a comparison table of LeanGeo with other Lean benchmarks mentioned in the paper.
>
> | Benchmark            | Size | Verifiable | Geometric Formal Proving Percentage | Lean | Theorem Library | Level          |
> | -------------------- | ---- | ---------- | ----------------------------------- | ---- | --------------- | -------------- |
> | miniF2F              | 488  | ✓          | 0%                                  | ✓    | Mathlib         | Middle  School |
> | Geometry3K-test      | 601  | ✓          | 0%                                  |      |                 | Middle School  |
> | LeanEuclid [32]      | 173  | ✓          | 0%                                  | ✓    | SystemE         | Elementary     |
> | AlphaGeometry-IMO_30 | 30   |            | 100%                                |      | DD rules        | Olympiad-level |
> | MATP-Bench           | 1056 | ✓          | About 20%                           | ✓    |                 | Mixed-level     |
> | **LeanGeo-Bench**    | 123  | ✓          | 100%                                | ✓    | LeanGeo         | Mixed_level      |
>
> Across existing geometry or Lean benchmarks, **none simultaneously offer Olympiad-level difficulty, fully verifiable proofs, and a complete synthetic theorem library**. As shown in the Table, datasets such as miniF2F and Geometry3K do not contain geometric proving tasks; LeanEuclid remains restricted to elementary-school geometry and autoformalization but not theorem proving;  Alphageometry_IMO_30 is established on an isolated and inverifiable formal system; MATP-Bench supplies many problems but no theorem-library tools, leaving its tasks essentially unsolvable for LLM provers. **LeanGeo-Bench fills this gap for the first time**: it provides a *fully verifiable*, *100% geometrically provable*, *Lean-native* synthetic benchmark equipped with a dedicated theorem library constructed for automated reasoning.
>
> Regarding the request for a shared-subset comparison, we appreciate the suggestion; however, existing geometry benchmarks differ substantially in problem sources, evaluation protocols, data formats, and even underlying axiom systems. These incompatibilities make it impossible to construct a meaningful shared subset for side-by-side evaluation. Consequently, a directly aligned comparison is not feasible. For reference, we have included in the **Appendix E** several comparisons with AlphaGeometry to illustrate representative differences.
>
> **Q2: Scability of LeanGeo**
>
> *"Scalability and complexity are uncharacterized (no curves vs. points/constraints/branching)"*
>
> **Response 2.1**:
>
> We present supplementary experiments and detailed discussions about the scalability of LeanGeo in our paper, which  specifically investigates the impact of the following factors on compilation time and the number of heartbeats required for proof execution:
>
> 1. The number of basic geometry elements (i.e., the number of points, lines, and circles).
> 2. The number of conditions.
> 3. The length of the proof.
> 4. The number of uses of the **euclid tactics**.
>
> （See detailed scaling graphs in Appendix A.1)
>
> Across all four experiments, LeanGeo exhibits approximately linear scaling: as we increase assumptions, conditions, proof length, or the number of Euclid tactics, both heartbeats and compilation time grow in a strongly correlated, near-linear manner. The only noticeable rises occur when the logical structure becomes denser (e.g., deeper lemma dependencies), which naturally increases the amount of proof search. Overall, the results show that LeanGeo is practically scalable, with performance determined primarily by the expected positive correlation between heartbeats and compilation effort rather than by any pathological geometric cases.

---

> ### Author Response · Authors · 2025-11-20
> **Response to Reviewer JjNZ Part 2/2**
>
> **Q2: Scability of LeanGeo**
>
> *"What are scaling curves for success/time vs. theorem-library size and lemma granularity?"*
>
> **Response 2.2**
>
> Our experiments in **Appendix A.2** demonstrate that **coarse lemma granularity** leads to severe blow-ups in both compilation time and heartbeats. When large “all-in-one” lemmas are inlined directly into a theorem, many nearly identical reasoning steps must be recompiled repeatedly, causing exponential-like scaling.
>
> In contrast, extracting commonly reused intermediate results into separate lemmas keeps the compilation cost close to linear in the dependency depth, because each lemma is compiled once and then reused. This is precisely why Lean’s modular proof structure is essential for scalability.
>
> The table below illustrates this effect using Miquel’s Theorem at different lemma-dependency depths:
>
> | Lemma Depth | Compiled One Time (Hearbeats) | Compiled One Time (Compilation Time) | Compiled Multiple Times (Hearbeats) | Compiled Multiple Times (Compilation Time) |
> | :---: | :------: | :---------: | :------: | :---------: |
> | 0 | $1.7×10^5$ | $2.1×10^4ms$ | $1.7×10^5$ | $2.1×10^4ms$ |
> | 1 | $6.3×10^5$ | $8.8×10^4ms$ | $7.6×10^5$ | $1.0×10^5ms$ |
> | 2 | $1.4×10^6$ | $2.1×10^5ms$ | $2.7×10^6$ | $4.0×10^5ms$ |
> | 3 | $2.5×10^6$ | $3.6×10^5ms$ | $8.9×10^6$ | $1.3×10^6ms$ |
> | 5 | $4.6×10^6$ | $7.5×10^5ms$ | $1.0×10^8$ | $1.8×10^7ms$ |
> | 8 | $7.5×10^6$ | $1.3×10^6ms$ | $1.4×10^9$ | $2.2×10^8ms$ |
>
> We define lemma depth as the maximum dependency depth from the current theorem; depth 0 refers to the theorem itself. The left columns report the cost when intermediate lemmas are extracted and compiled once; the right columns show the cost when those lemmas are inlined and thus recompiled multiple times. As lemma depth grows, the scaling gap becomes dramatic—confirming that proper lemma granularity is crucial for maintaining manageable compilation cost.
>
> **Q3: SMT-off experiments**
> *"How does performance change with SMT off or with restricted solver capabilities?"*
> **Response**
> We are unable to conduct the requested ablation (disabling SMT or restricting solver capabilities), but we emphasize that SMT plays a crucial role in our theorem-library construction. Its function is analogous to—but substantially more powerful and structurally sophisticated than—Lean’s high-level automation such as `nlinarith` or `ring`. Without SMT, the proofs in our library would become significantly more cumbersome, often requiring long manual derivations. As a reference point, Beeson et al. (2019) formalized Euclid’s *Elements* in Coq **without SMT**, and even the very first proposition required nearly 100 lines of highly formal, low-level reasoning that diverges from Euclid’s original structure. In contrast, **with SMT support our formalization of the same result takes fewer than 10 lines**, illustrating the substantial efficiency and practicality SMT brings to synthetic geometry in LeanGeo.
>
> We again thank the reviewer for the valuable feedback. The comments have significantly improved the clarity and positioning of the paper. Let us know if additional clarifications are needed.

---

> ### Comment · Area_Chair_XcaY · 2025-11-28
>
> Dear Reviewer JjNZ,
>
> The authors gave their rebuttals to your reviews. Could you please express your opinions?
>
> Many thanks,
> AC

---

### Official Review · Reviewer_hrZo · 2025-11-01

**Soundness:** 2
**Presentation:** 2
**Contribution:** 3
**Rating:** 6
**Confidence:** 3

**Summary:**

This paper introduces LeoGeo, a formal system for formulating and solving competition-level geometry problem, and LeoGeo-Bench, a collection of benchmark geometry problems, both built using Lean 4. Benchmark results for state-of-the-art LLMs are provided.

**Strengths:**

* The LeanGeo library allows formalizing and solving competition-level geometry problems in Lean 4. It includes an extensive library of high-level definitions and tactics, which makes formal proofs more intuitive and understandable. LeanGeo's integration with Mathlib allows it to leverage powerful tools from other areas of maths.
* The LeanGeo-Bench benchmark is useful for evaluating advances in the field.
* The paper is generally well-written and easy to follow.

**Weaknesses:**

I find some of the claims insufficiently supported and some more details would be helpful, as explained below.
* The main limitation of LeanEuclid compared to LeanGeo seems to be a limited set of formalized geometry facts, as stated at line 52. Is it difficult to expand LeanEuclid's library? If not, what additional advantage does LeanGeo has?
* The paper claims that LeanGeo allows expressing and reasoning about geometry problems in a human-like manner. This may be debatable as the examples presented in the paper are still highly formal. It is also not clear what exactly is done to make the proofs more "human-like". Can you provide more details?
* At some places the writing could be improved. The RL experiments seem to be a significant part of the paper, but this is not motivated in the introduction, and not mentioned in the abstract and the list of contributions.

Minor
* Line 234: Line 1 should pass the point B, but this is not mentioned.
* Line 820: empty bullet point.

**Questions:**

See questions in Weaknesses and the questions below.

* Is mathlib always used for generating proofs using the theorem prover? If yes, is it possible to evaluate the theorem prover's performance without using it?
* Only 43 problems in the benchmark have proofs. Is it because the automatic theorem prover fails to provide proofs for other problems? If yes, on which problems the theorem prover fails?

---

> ### Author Response · Authors · 2025-11-20
> **Response to Reviewer hrZo 1/2**
>
> We sincerely thank the reviewer for the thoughtful and constructive comments. We address each weakness and question below, and we have revised the paper accordingly.
>
> **W1. Comparison to LeanEuclid and difficulty of expanding its library**
>
> *“Is it difficult to expand LeanEuclid’s library? If not, what additional advantage does LeanGeo have?”*
>
> **Response**
>
> The core contribution of LeanGeo is precisely the **substantial expansion** of LeanEuclid’s theorem library and geometric structures. LeanEuclid formalizes only the 49 propositions in Euclid’s Elements I; consequently, its expressiveness is far from sufficient for solving standard middle- and high-school geometry problems. Below is a summary comparison included in the revised version of the paper:
>
> |                                             | LeanEuclid       | LeanGeo                |
> | ------------------------------------------- | ---------------- | ---------------------- |
> | Axiom Number                                | 107              | 116                    |
> | Theorem Number                              | 106              | 260                    |
> | Geometry Structure Number                   | 12               | 50                     |
> | Average Proof Length                        | 20.27            | 16.20                  |
> | Average number of quote lemma               | 3.80             | 3.43                   |
> | Average Proof Length(Using LeanEuclid only) | 20.27            | 1562.1                 |
> | SMT                                         | Hard-coded       | LeanSMT                |
> | Level                                       | Euclid's Element | Competitional Geometry |
>
> LeanGeo **slightly modifies LeanEuclid’s axiom system** and we **build a much richer theorem library and structure ecosystem** on top of it. These higher-level tools drastically shorten proofs: if one attempts to solve LeanGeo-level problems using only LeanEuclid’s primitive theorems, the proofs are still theoretically possible, but the average length exceeds 1500 lines. Moreover, LeanGeo is substantially **more extensible** than LeanEuclid: our SMT translation layer is fully flexible and can accommodate newly introduced axioms, constants, and definitions, whereas LeanEuclid’s SMT interface is largely hard-coded and difficult to extend.
>
> Expanding the theorem library is **nontrivial**. The development of our theorem base required two CMO gold medalists working full-time for two months. The challenge is that each result must be proven rigorously and in a graph-free, fully formal style. Many statements that are intuitively obvious become extremely technical to formalize. For instance, formally proving the existence of a triangle’s incenter is surprisingly delicate without diagrammatic intuition: in Lean, our proof spans 32 lines and invokes 18 lemmas, even though in natural language this fact requires virtually no justification at all. We devoted substantial effort to overcoming such obstacles and formalizing results that future users can rely on directly—providing the foundational infrastructure that bridges Euclidean axioms to competition-level geometry, allowing subsequent researchers to develop Lean-based geometry far more efficiently.
>
> **W2. “Human-like manner” in LeanGeo**
>
>  *“The examples are still highly formal, what exactly makes the proofs more ‘human-like’?”*
>
> **Response:**
>
> Thank you for raising this important point. We agree that Lean’s syntax is inherently formal and cannot resemble natural human writing. The term *“human-like manner”* in the original draft was indeed not ideal, and we have refined the wording in the revised version.
>
> What we intended to convey is that **LeanGeo’s proof structure and reasoning patterns more closely mirror human geometric reasoning**, especially when compared to existing Lean-based tools or search-driven systems like AlphaGeometry. This resemblance appears in several dimensions:
>
> 1. **Hierarchical, intention-revealing decomposition via `have`**
>    LeanGeo encourages breaking down proofs into human-recognizable subgoals. For example, in IMO 2000 P1, human solvers typically identify milestones such as establishing that T is the midpoint of AB, proving △ABE≅△ABM. LeanGeo mirrors this structure: each conceptual step becomes a top-level `have`, and each `have` is further decomposed using geometric identities (e.g., ratio lemmas, intercept theorems).
>
> 2. **Human-like invocation of advanced theorems**
>    Human solvers routinely use classical theorems—intercept theorems, intersecting chords, parallel line lemmas—rather than reconstructing them from primitive axioms.
>
>    LeanGeo formalizes such high-level theorems and makes them callable. In IMO 2000 P1, we invoke the cutting-line theorem, the intersecting-chords theorem, and the parallel-line segment lemma to prove the theorems.

---

> ### Author Response · Authors · 2025-11-20
> **Response to Reviewer hrZo Part 2/2**
>
> **Q1:Importance of Theorem Library in automated theorem proving**
>
> *“Is mathlib always used for generating proofs using the theorem prover? If yes, is it possible to evaluate the theorem prover's performance without using it?”*
>
> **Response:**
>
> In practice, **no**. State-of-the-art theorem provers depend heavily on Mathlib. Without Mathlib, models lose access to even the simplest tools (e.g., proving 1>0), and performance collapses to near zero on any benchmark. Similarly, without LeanGeo, models have effectively no ability to solve formal geometry problems, because even basic tools for similarity, congruence, or classical lemmas are unavailable.
>
> Thus, evaluating theorem provers without LeanGeo may fail to capture their true geometric-reasoning capabilities, since they would be forced to re-derive numerous basic facts from scratch, which is not consistent with standard evaluation practice.
>
> We note that many general-purpose LLMs (Gemini, GPT, etc.) have been trained on formal mathematics datasets including Mathlib, enabling them to use these libraries. However, LeanGeo is new: **no existing prover or LLM has prior training on Lean-style formal geometry**, and our goal is precisely to bridge this gap.
>
> **Q2: Why the LLMs fails on LeanGeo-bench**
>
> *"Only 43 problems in the benchmark have proofs. Is it because the automatic theorem prover fails to provide proofs for other problems? If yes, on which problems the theorem prover fails?"*
>
> **Response:**
>
> As stated in the paper, the 43 provided proofs are **human-written reference solutions**. These correspond to all 40 problems in UG, LB, and SP,1 HSC problem, and 2 IMO problems.
> Beyond these, LLMs were able to generate correct proofs for an additional 7 HSC problems.
>
> Table 3 in the paper documents model performance. In particular, the models **fail on all OP and IMO problems**.
> This failure has two primary causes:
>
> 1. **Weak geometry problem-solving ability** of current general LLMs, which often rely on coordinate-geometry heuristics or special-case reasoning rather than competition-style synthetic geometry.
>
> 2. **The difficulty of conveying LeanGeo proof-style knowledge and lemma usage via prompting alone**. Many of the key technical tricks used to construct our theorem library are hard to describe textually and hard for LLMs to learn without explicit training.
>
> In short, the limitations arise not because the problems are unformulatable, but because **LLMs currently lack sufficiently strong geometric and formal LeanGeo reasoning abilities**, and the theorem library (while substantial) still needs further expansion.
>
> ------
>
> **We again thank the reviewer for the valuable feedback.** The comments have significantly improved the clarity and positioning of the paper. Let us know if additional clarifications are needed.

---

> > ### Comment · Area_Chair_XcaY · 2025-11-28
> >
> > Dear Reviewers,
> >
> > How do you think about the authors' rebuttals?
> >
> > Best wishes,
> > AC

---

### Author Response · Authors · 2025-11-20
**Summary of Revisions to the LeanGeo Paper**

We sincerely thank all reviewers for their constructive feedback. Following the suggestions, we have made substantial revisions to improve the clarity, completeness, and contributions of the paper. **All modifications are highlighted in blue in the revised version.** The main updates are as follows:

------

### **1. Introduction and Abstract**

We have updated both the *introduction* and *abstract* to explicitly include **reinforcement training** as part of our contributions, clarifying LeanGeo’s learning pipeline and its significance to automated geometric reasoning.

------

### **2. Related Work**

To better highlight the value and contribution of **LeanGeo**, we reorganized and expanded the *Related Work* section by adding three explicit comparisons with existing formal geometry systems and benchmarks. These comparisons are summarized in **Tables 2, 3, and 4**.

#### **(1) Comparison with LeanEuclid (Table 2).**

We added a detailed comparison between **LeanGeo** and **LeanEuclid** (Murphy et al., 2024). While both systems share the same axiomatic foundation, **LeanGeo** provides a significantly expanded set of theorems, definitions, and geometric constructions, and incorporates a more powerful SMT-based reasoning mechanism.

#### **(2) Comparison with AlphaGeometry (Table 3).**

Table 3 contrasts **LeanGeo** with **AlphaGeometry** (Trinh et al., 2024) in terms of expressive power, verifiability, and axiomatics. We also provide clarifying explanations to delineate the conceptual and methodological differences between the two systems.

#### **(3) Comparison with Other Geometry Benchmarks (Section 2.4 & Table 4).**

We added **Section 2.4** to survey additional geometry-related and Lean-related benchmarks.
 **Table 4** presents a comparison between **LeanGeo-Bench** and existing geometry benchmark suites, highlighting differences in scale, structural diversity, and formal verification support.

------

### **3. LeanGeo Section**

We added two paragraphs explaining the **scalability of LeanGeo**, focusing on lemma granularity, geometric elements, conditions, proof length, and the use of Euclid tactics. These discussions are based on the findings detailed in Appendix A. We also fixed a minor mistake in Figure 2.

------

### **4. Appendix Revisions**

#### **Appendix A.1 – Scalability Analysis**

We now provide a comprehensive scalability evaluation showing that LeanGeo’s compilation time and heartbeats grow in a stable, near-linear manner across variations in geometric elements, assumptions, proof lengths, and uses of Euclid tactics.

#### **Appendix A.2 – Importance of Lemma Granularity**

We demonstrate that appropriate lemma granularity significantly impacts efficiency: modularizing intermediate results maintains linear complexity with respect to dependency depth, whereas coarse “all-in-one” proofs may lead to exponential-style blow-ups.

#### **Appendix E – Case Studies Comparing LeanGeo and AlphaGeometry**

We present 8 case comparison to systematically compares the expressive power and logical soundness of LeanGeo and AlphaGeometry, showing that more than half of LeanGeo’s theorems cannot even be stated in AlphaGeometry due to its restricted formal language. We further demonstrate that AlphaGeometry’s diagram-dependent inference rules can produce contradictory conclusions under identical assumptions, underscoring the necessity of LeanGeo’s fully axiomatized, diagram-free framework for reliable and generalizable geometric reasoning.

---

### Author Response · Authors · 2025-12-02
**Summarizing the Contributions of this Paper**

Dear Reviewers and ACs,

Thank you for your time and effort in reviewing our paper. We are grateful for the positive recognition from all reviewers. Below we provide a concise summary of our paper’s main contributions:

1. We present **LeanGeo**, the first framework within the Lean theorem prover capable of expressing and reasoning about competition-level geometry problems. This formal system is built upon the rigorous axiomatic foundation and places a stronger emphasis on **logical rigor and completeness**. Our system offers a meaningful foundation that supports the continued development of Euclidean geometry formalization in Lean 4.

2. We introduce a comprehensive geometry benchmark formalized in Lean 4 and LeanGeo, capable of representing the majority of International Mathematical Olympiad (IMO) geometry problems. Our benchmark is distinguished by its dedicated focus on the **logical verification of formal proofs**. Lean and LeanGeo provide rich **geometric reasoning tools** for our evaluation platform, which supports truly rigorous, machine-verified assessments of planar geometry proofs.

3. We develop a novel method for generating synthetic competition-level geometry problems, together with a reinforcement learning pipeline designed to equip LLMs with previously unseen geometric knowledge.

Here we also summarize the main explanations provided in the rebuttal and the revised paper:

1. Through both case studies and quantitative analyses, we compare LeanGeo and LeanGeo-bench with existing geometry frameworks (e.g., AlphaGeometry, LeanEuclid) and benchmarks (e.g., MATP-bench), underscoring our focus on rigorously formalizing difficult planar-geometry proofs.

2. We further include new experiments on the **scalability** of LeanGeo, demonstrating the advantages of our lemma-retrieval architecture for geometric automated theorem proving.

---

### Meta-Review · Area_Chair_t4Ks · 2025-12-23

**Summary:**

This paper proposes LeanGeo, a Lean 4 based framework for formalizing competition level Euclidean geometry, together with LeanGeo Bench (122 problems) and an initial synthetic data plus RL training pipeline. Reviewers agree the overall direction is useful and the library plus benchmark could become valuable infrastructure for formal geometry and LLM theorem proving in Lean.

The main concerns are that the **core novelty** is largely an expansion and engineering integration on top of prior Lean geometry efforts (for example LeanEuclid and existing axiomatics), while the experimental section is thin and not diagnostic. In particular, reviewers found **limited ablations**, **unclear attribution of failures**, and a **preliminary RL component** without sufficient methodological detail, baselines, or analysis.

There are also concerns about **benchmark scale** and the **lack of truly aligned side by side comparisons** on shared subsets with existing systems, leaving some positioning claims insufficiently supported. Overall these issues led reviewers to lean below the acceptance threshold.

**Reviewer Concerns:**

Addressed in the rebuttal and revision:

- The paper now better motivates and situates the RL component, and clarifies it as preliminary.
- The authors provided clearer comparisons to LeanEuclid and other benchmarks via new tables, and added discussion and additional evidence on scalability (including lemma granularity effects).
- The authors refined the overstatement around “human like” proofs, clarifying that the intent is higher level decomposition and use of classical theorems rather than natural language similarity.
- Clarifications were provided on why only a subset has human written reference proofs, and where current LLMs fail by problem category.

Still outstanding:

- The experimental evaluation remains insufficiently diagnostic for the central claims. Key ablations requested by reviewers are still missing, most notably SMT off or restricted SMT settings, plus controlled studies of tactic schedules, retrieval settings, library size, and prompting or decoding.
- Baseline comparisons to strong Lean provers or geometry solvers remain limited, and the rebuttal relies partly on feasibility arguments rather than evidence.
- Novelty concerns remain for at least two reviewers: the contribution is still perceived as primarily library expansion plus integration, with limited conceptual or algorithmic innovation beyond infrastructure work.
- Benchmark scale and representativeness concerns are only partially addressed. The cost based rationale is reasonable, but the community value and coverage relative to larger suites is still not fully demonstrated empirically.
- **There is an authorship anonymity concern raised in discussion**, which does not change the technical assessment but is a process issue.

**Reviewer Scores:**

- Reviewer hrZo (6): Likely stays at 6. The rebuttal addresses most of the specific questions, but does not materially change the strength of evidence on the main experimental limitations.
- Reviewer JjNZ (4): Likely stays at 4. Added benchmark comparisons and scalability discussion help, but the core critique about missing ablations and non diagnostic experiments remains, and the inability to run SMT off ablations is directly at odds with the reviewer’s request.
- Reviewer iGd7 (4): Likely stays at 4. The rebuttal clarifies positioning versus Myers and provides qualitative arguments against AlphaGeometry style formalisms, and explains benchmark size constraints, but the novelty and RL depth concerns are not fully resolved, and quantitative aligned comparisons and stronger baselines are still missing.

---

### Decision · Program_Chairs · 2026-01-26

Reject